# Dual heterogeneous interfaces enhance X-ray excited persistent luminescence for low-dose 3D imaging

Lei Lei [1,3] ✉, Minghao Yi[1,3], Yubin Wang[1], Youjie Hua[1], Junjie Zhang [1], Paras N. Prasad [2] ✉ & Shiqing Xu [1] ✉

Lanthanide-doped fluoride nanoparticles (NPs) showcase adjustable X-ray-excited persistent luminescence (XEPL), holding significant promise for applications in three-dimensional (3D) imaging through the creation of flexible X-ray detectors. However, a dangerous high X-ray irradiation dose rate and complicated heating procedure are required to generate efficient XEPL for high-resolution 3D imaging, which is attributed to a lack of strategies to significantly enhance the XEPL intensity. Here we report that the XEPL intensity of a series of lanthanide activators (Dy, Pr, Er, Tm, Gd, Tb) is greatly improved by constructing dual heterogeneous interfaces in a double-shell nanostructure. Mechanistic studies indicate that the employed core@shell@shell structure could not only passivate the surface quenchers to lower the non-radiative relaxation possibility, but also reduce the interfacial Frenkel defect formation energy leading to increase the trap concentration. By employing a NPs containing flexible film as the scintillation screen, the inside 3D electrical structure of a watch was clearly achieved based on the delayed XEPL imaging and 3D reconstruction procedure. We foresee that these findings will promote the development of advanced X-ray activated persistent fluoride NPs and offer opportunities for safer and more efficient X-ray imaging techniques in a number of scientific and practical areas.

Scintillation-based X-ray imaging technology has found extensive applications in medical diagnostics[1–4], security screening[5,6], astronomical discovery[7,8] and industrial inspection[9,10]. Unlike commercial thin-film transistors integrated flat-panel X-ray detectors, the emerging flexible and thus conformable X-ray detectors offer the advantage of three-dimensional (3D) imaging of curved or irregular objects[11]. X-ray excited persistent luminescence (XEPL) materials, such as lanthanide-doped fluoride nanoparticles (NPs) and halide perovskites, are promising candidates for the fabrication of flexible X-ray detectors[12–15]. Lanthanide-doped fluoride NPs, with their covalent crystal structures, exhibit superior stability compared to halide perovskites with ionic crystal structures. Additionally, designing fluoride-based core/shell nanostructures with ease to manipulate excitation dynamics offers the ability to control XEPL properties. For instance, a shell layer can passivate surface quenchers, resulting in enhanced XEPL intensity[16–19], while spatial separation of activators in the core and the shell layer allows for time-dependent XEPL color evolutions[20]. Despite some progress being made in manipulating XEPL of lanthanide-doped fluoride NPs, achieving a significant increase in the XEPL brightness through the design of advanced nanostructures remains a major

[1]Key Laboratory of Rare Earth Optoelectronic Materials and Devices of Zhejiang Province, Institute of Optoelectronic Materials and Devices, China Jiliang University, Hangzhou 310018, P.R. China. [2]Institute for Lasers, Photonics, and Biophotonics and Department of Chemistry, University at Buffalo, State University of New York, Buffalo, NY 14260, USA. [3]These authors contributed equally: Lei Lei, Minghao Yi. ✉e-mail: leilei@cjlu.edu.cn; pnprasad@buffalo.edu; shiqingxu@cjlu.edu.cn

challenge. This advancement is crucial for reducing potentially harmful X-ray irradiation dose rates and eliminating the need for complex thermal processing.

XEPL in lanthanide-doped fluoride NPs originates from the release of deposited electrons in the traps formed by the F⁻ related Frenkel defects after the cessation of X-rays[13,20]. As schematically illustrated in Fig. 1a, small fluoride NPs possess a large surface-to-volume ratio, and many surface F⁻ ions are incompletely coordinated. Consequently, X-ray irradiation induces the formation of Frenkel defects predominantly on the NP surface. While a general core@shell structure can reduce energy migration efficiency from activators to surface quenchers[20], they also inhibit the formation of Frenkel defects due to the recovery of the surface F⁻ ions coordination environment[21] (Supplementary Fig. 1 and Supplementary Note 1). Therefore, it is hard to greatly improve the XEPL intensity by coating a normal shell layer. Introducing a strain at the core-shell interface arising from lattice mismatches[22], has proven effective in engineering heterostructures to optimize properties such as photoluminescence[23,24], catalytic activity[25,26], and magnetism[27,28]. From this view of point, engineering the interfacial misfit strain probably serves as an effective approach to

reduce the formation energy of Frenkel defects in interfacial F⁻ ions and subsequently amplify the XEPL intensity.

In this work, we present a significant improvement in the XEPL intensity achieved by constructing a dual heterogeneous interface. The employed core@shell@shell structure serves two purposes: passivating the surface quenchers to reduce non-radiative relaxation possibilities, and lowering the interfacial Frenkel defect formation energy, leading to increased trap concentration due to the presence of misfit strain at the heterogeneous interfaces (Fig. 1a). By restricting the Dy activators to the interfacial layer of the Y@Lu/Gd/Dy@Y heterogeneous core@shell@shell NPs, we observed a 40.9-fold enhancement in XEPL intensity compared to that of the general Lu/Gd/Dy core NPs. Similar enhancements in the XEPL intensity were observed for other lanthanide activators (Pr, Er, Tm, Gd, Tb) when employing a similar core@shell@shell structure. Additionally, the XEPL durations of these designed NPs were significantly prolonged. Using the Y@Lu/Gd/Tb@Y NPs incorporated into a flexible film as a scintillation screen, we recorded the internal electrical structures of a watch from five directions using high-resolution delayed XEPL imaging. Subsequently, a 3D reconstruction procedure enabled the visualization of its stereoscopic structure.

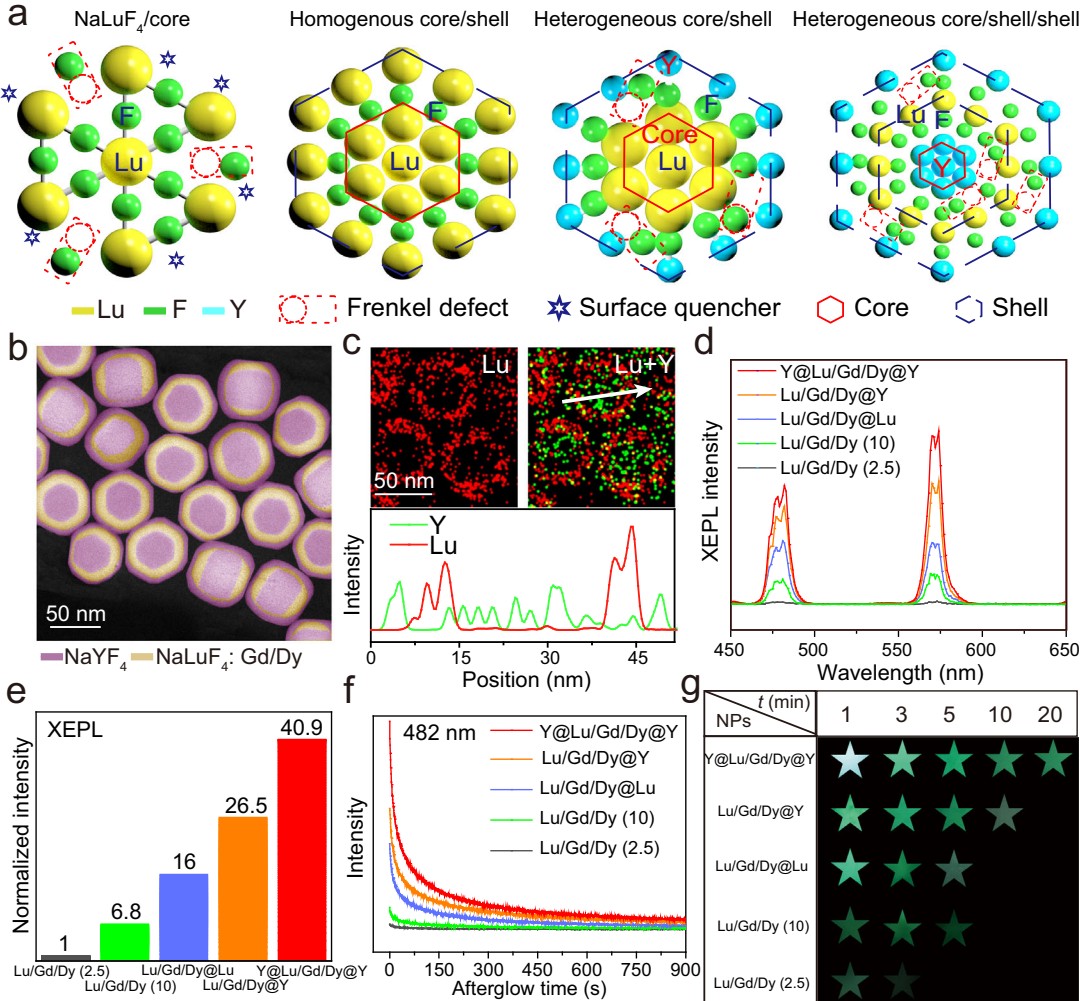

**Fig. 1 | Compared XEPL properties of Dy doped fluoride NPs with different structures. a** Schematic illustration for the influence of shell layer on the passivation of surface quenchers and the generation of Frenkel defects. The core NP contains many surface quenchers, and the incompletely coordinated surface F⁻ ions facilitate the formation of Frenkel defects upon X-ray irradiation. The inert shell serves to passivate the surface quenchers. A homogenous core@shell structure hinders Frenkel defects formation, while the heterogeneous core@shell structure with a heterogeneous interface promotes it. HAADF image (**b**), EDX line scan and mapping results (**c**) of the Y@Lu/Gd/Dy@Y NPs. The line scan is along the direction indicated by the white arrow. Red, Lu signal; green, Y signal. XEPL spectra (**d**), integral XEPL intensities (**e**), XEPL decay curves (**f**), and digital photographs (**g**) of the studied NPs at different time delays. Source data are provided as a Source Data file.

## Results

### Spectroscopic study of the enhancement of XEPL

As shown in Supplementary Fig. 2 and Table 1, the mean particle sizes of the Lu/Gd/Dy ([Na]/[RE] = 2.5), Lu/Gd/Dy ([Na]/[RE] = 10), Lu/Gd/Dy@Lu, Lu/Gd/Dy@Y, Y@Lu/Gd/Dy@Y NPs were about 35 nm, 36 nm, 64 nm, 62 nm and 65 nm, respectively. A high-resolution transmission electron microscopy (TEM) image revealed the single crystal nature of an individual NP, and the observed clear lattice fringes with a d-spacing of 0.521 nm were assigned to the (100) plane (Supplementary Fig. 2h). All these NPs were well indexed to a pure hexagonal phase without any impurity peaks (Supplementary Fig. 3). Energy dispersion X-ray spectroscopy (EDX) analysis confirmed the presence of Lu, Gd, Dy and F elements in the NPs (Supplementary Fig. 4 and Supplementary Table 1). The high-angle annular dark-field (HAADF) images, EDX line scans and element mapping results confirmed the formation of Lu/Gd/Dy@Lu (Supplementary Fig. 5) and Lu/Gd/Dy@Y (Supplementary Fig. 6) core@shell as well as Y@Lu/Gd/Dy@Y (Fig. 1b and Supplementary Figs. 7, 8) core@shell@shell structures. As revealed in Fig. 1c, the Lu and Gd were primarily distributed in the interlayer and Y was located in the core and the outer shell, which were consistent with the nominal compositions of the core@shell@shell structure. Increasing the [Na]/[RE] ratio in the precursor solution from 2.5 to 10 resulted in a 6.8-fold increase in the XEPL intensity (Fig. 1d, e). Unless otherwise specified, we prepared the emissive and inert layers using [Na]/[RE] ratios of 10 and 2.5, respectively. Coating an inert shell of $NaLuF_4$ or $NaYF_4$ to block the energy migration from activators to surface quenchers led to an enhanced XEPL intensity for the Dy activators (Fig. 1d). Interestingly, the XEPL intensity of the Lu/Gd/Dy@Y NPs was much stronger than that of the Lu/Gd/Dy@Lu NPs with similar core sizes and shell thicknesses, indicating that the heterogeneous interface produced an improvement in the XEPL intensity. As anticipated, the introduction of one more heterogeneous interface by constructing a sandwich structure of Y@Lu/Gd/Dy@Y further enhanced the XEPL intensity (Fig. 1d). Specifically, the XEPL intensity of the Y@Lu/Gd/Dy@Y NPs was approximately 40.9 times higher than that of general Lu/Gd/Dy ([Na]/[RE] = 2.5) core NPs (Fig. 1e). The optimal Dy doping concentration in the Y@Lu/Gd/Dy@Y NPs was measured to be ≈2 mol % (Supplementary Fig. 9), and the XEPL intensity enhancement phenomenon was independence of the Dy doping concentration (Supplementary Fig. 10). The XEPL decay curves further demonstrated that both the intensity and the duration were significantly improved by incorporating the Dy activators into the interlayer of the Y@Lu/Gd/Dy@Y core@shell@shell NPs (Fig. 1f). As revealed by the photographs (Fig. 1g), the bright XEPL emitted by the Y@Lu/Gd/Dy@Y NPs remained visible even after 20 min, while that of Lu/Gd/Dy ([Na]/[RE] = 2.5) core NPs became very weak within only 3 min.

### Confirming the enhancement of XEPL by designing comparable nanostructures

To further confirm the improvement of XEPL intensity through the construction of a heterogeneous interface, the Lu@Lu/Dy@Lu, Lu@Lu/Dy@Y and Y@Lu/Dy@Y NPs with similar mean particle sizes were prepared and studied (Fig. 2a–f and Supplementary Figs. 11, 12). It is important to note that the Gd ions were not incorporated in the interlayer to avoid their influences on the XEPL intensity variations[13,20]. The interlayer thickness in these core@shell@shell NPs were about 5 nm (Table 1). As shown in Fig. 2g, the XEPL intensity of the Y@Lu/Dy@Y NPs was stronger than that of Lu@Lu/Dy@Y NPs, and both were stronger than that of Lu@Lu/Dy@Lu NPs. The XEPL decay curves further confirmed that the Y@Lu/Dy@Y NPs with a dual heterogeneous interface exhibited the highest XEPL intensity and the longest duration (Supplementary Fig. 13). Similar trends in XEPL intensity variation were observed when the interlayer thickness was increased from 5 nm to 9 nm (Fig. 2h and Supplementary Figs. 14, 15). Interestingly, even with the incorporation of $Gd^{3+}$ ions in the interfacial layer,

the Y@Lu/Gd/Dy@Y NPs continued to exhibit the highest XEPL intensity (Supplementary Figs. 16–18). As expected, the Y@Lu/Tb@Y NPs with dual heterogeneous interfaces showed stronger XEPL intensity and longer XEPL duration compared to the Lu@Lu/Tb@Y and Lu@Lu/Tb@Lu NPs (Fig. 2i and Supplementary Fig. 19). Despite the possible presence of cation intermixing during shell growth[24,29,30], the XEPL intensity of core@shell NPs with a heterogeneous structure remained stronger than that of their homogeneous counterparts (Supplementary Fig. 20 and Supplementary Note 2). These results provide further evidence that a heterogeneous core@shell@shell structure enhances the XEPL performance of lanthanide activators.

### Mechanistic investigation

To gain a deeper understanding of the mechanism behind the enhancement of XEPL intensity produced by a heterogeneous interface, additional analysis and theoretical calculations were performed. Density functional theory (DFT) was used to calculate the formation energies ($E_f$) of anion Frenkel defects, which represent the energy required for the dislocation of interfacial $F^-$ ions into interstitial sites with different separation distances (0.2, 0.5, 0.7, 1.0, 1.5, and 2.0 Å) in homogeneous (Lu@Lu, Y@Y) and heterogeneous (Lu@Y) core@shell structures (Fig. 3a and Supplementary Fig. 21). As shown in Fig. 3b, the $E_f$ values for the heterogeneous case were smaller compared to the homogeneous case, especially when the separation distance exceeded 0.5 Å, indicating that a heterogeneous interface facilitates the formation of anion Frenkel defects through elastic collisions between large-momentum X-ray photons and small fluoride ions. X-ray activated thermoluminescence (TL) spectra were utilized to investigate the variations in trap density. As shown in Fig. 3c, the TL intensity of the Y@Lu/Dy@Y NPs was much stronger than that of Lu@Lu/Dy@Y NPs, and both were superior to that of Lu@Lu/Dy@Lu NPs. After the cessation of X-ray excitation, the electronic paramagnetic resonance (EPR) signal intensity ($g = 1.8743$, $g = 2.1163$) was stronger in the Y@Lu/Dy@Y NPs compared to the Lu@Lu/Dy@Lu counterparts (Supplementary Fig. 22). The binding energy of the F1s decreased after the formation of the heterogeneous interface (Supplementary Fig. 23). These experimental findings demonstrate that the construction of a heterogeneous core@shell structure promotes the generation of traps upon X-ray irradiation, which is consistent with the results obtained from the DFT calculations.

To confirm the presence of lattice distortion in the heterogeneous interface, a fast Fourier transform (FFT) pattern analysis was conducted. The FFT pattern revealed two distinct sets of diffraction spots corresponding to the core and the interlayer of a single NP, indicating a difference in lattice constants between the two regions (Fig. 3d). The distinctions observed in the slow-scan X-ray diffraction (XRD) patterns between the Y@Y/Dy@Y and Y@Lu/Dy@Y NPs (Supplementary Fig. 24) manifest substantial variations in interplanar spacing values and their corresponding ratios between the $NaYF_4$ and $NaLuF_4$ crystals. These observations suggest the presence of lattice distortion at the interface[31]. A geometric phase analysis of the high-resolution TEM image of a single core@shell@shell NP was employed to visualize the strain distribution[22,32]. The false-color maps of relative d-spacing in the {110} plane clearly showed the presence of a misfit strain in the dual heterogeneous interface (Fig. 3e). In contrast, the d value remained relatively uniform in the unstrained core layer. Based on the above results, a possible mechanism for the enhanced XEPL intensity is proposed (Fig. 3f). When the X-ray photons interact with the heavy lanthanide ions, hot electrons and deep holes are generated primarily through the photoelectric effect. Subsequently, numerous secondary electrons with low kinetic energies are produced through electron-electron scattering and the Auger process. These secondary electrons can be captured by traps, leading to subsequent XEPL emission[33–36]. In comparison to the core NPs, the core@shell@shell NPs with a dual heterogeneous interface not only passivates the surface quenchers,

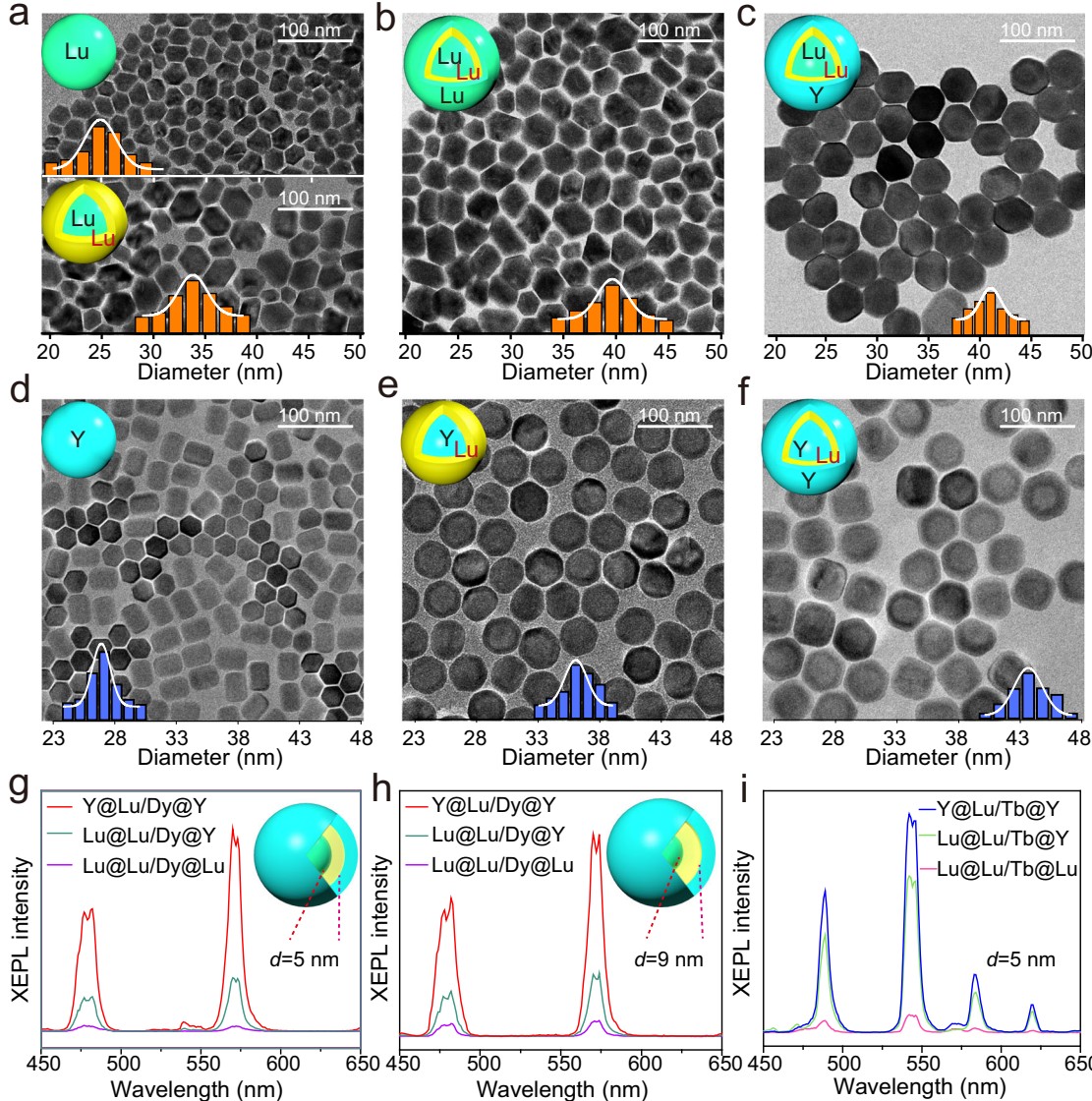

**Fig. 2 | Compared XEPL properties of lanthanide activators doped in the homogeneous and heterogeneous core@shell@shell structures.** TEM images of the NaLuF$_4$ (**a**, top), Lu@Lu/Dy (**a**, bottom), Lu@Lu/Dy@Lu (**b**), Lu@Lu/Dy@Y (**c**), NaYF$_4$ (**d**), Y@Lu/Dy (**e**), Y@Lu/Dy@Y (**f**). Green, NaLuF$_4$; yellow, NaLuF$_4$:Dy; blue, NaYF$_4$. Insets are histograms illustrating size distributions of their corresponding

40–100 NPs. XEPL spectra of the Y@Lu/Dy@Y (red), Lu@Lu/Dy@Y (green) and Lu@Lu/Dy@Lu (purple) core@shell@shell NPs with different interlayer thicknesses, 5 nm (**g**), 9 nm (**h**). **i** XEPL spectra of the Y@Lu/Tb@Y (blue), Lu@Lu/Tb@Y (green) and Lu@Lu/Tb@Lu (pink) core@shell@shell NPs. Source data are provided as a Source Data file.

**Table 1 | Measured sizes of the studied products with their corresponding abbreviations**

| Sample (abbreviation) | Size ([Na]/[RE]) | | | Total size |
|---|---|---|---|---|
| | **Core** | **First shell** | **Second shell** | |
| NaLuF$_4$:20Gd/2Dy, [Na]/[RE] = 2.5 (Lu/Gd/Dy (2.5)) | 35 nm (2.5) | / | / | 35 nm |
| NaLuF$_4$:20Gd/2Dy, [Na]/[RE] = 10 (Lu/Gd/Dy (10)) | 36 nm (10) | / | / | 36 nm |
| NaLuF$_4$:20Gd/2Dy@NaLuF$_4$ (Lu/Gd/Dy@Lu) | 36 nm (10) | 14 nm (2.5) | / | 64 nm |
| NaLuF$_4$:20Gd/2Dy@NaYF$_4$ (Lu/Gd/Dy@Y) | 36 nm (10) | 13 nm (2.5) | / | 62 nm |
| NaYF$_4$@NaLuF$_4$:20Gd/2Dy@NaYF$_4$ (Y@Lu/Gd/Dy@Y) | 35 nm (2.5) | 8 nm (10) | 7 nm (2.5) | 65 nm |
| NaLuF$_4$@NaLuF$_4$: 2Dy@NaLuF$_4$ (Lu@Lu/Dy@Lu) | 25 nm (2.5) | 4.5 nm (10) | 3 nm (2.5) | 40 nm |
| NaLuF$_4$@NaLuF$_4$: 2Dy@NaYF$_4$ (Lu@Lu/Dy@Y) | 25 nm (2.5) | 4.5 nm (10) | 3.5 nm (2.5) | 41 nm |
| NaYF$_4$@NaLuF$_4$: 2Dy@NaYF$_4$ (Y@Lu/Dy@Y) | 26 nm (2.5) | 5 nm (10) | 3.5 nm (2.5) | 43 nm |

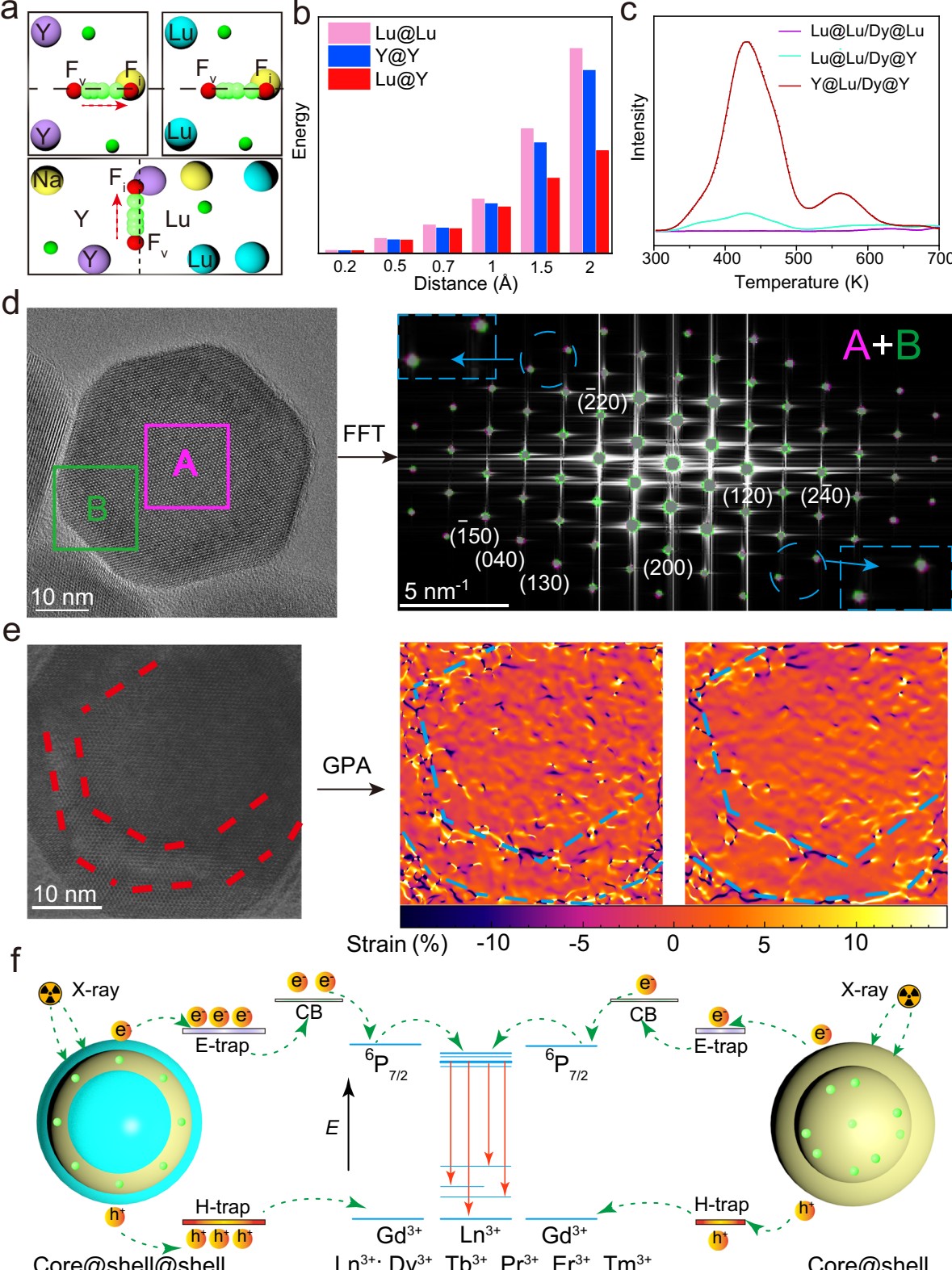

**Fig. 3 | Analysis of interfacial structure and energy transfer in the heterogeneous core@shell@shell NPs. a** Illustration of the dislocation of interfacial F⁻ ions into interstitial sites within the homogeneous (Lu@Lu, Y@Y) and heterogeneous (Lu@Y) core@shell structures. **b** Calculated Frenkel defect formation energies in these three core@shell NPs. **c** TL spectra of the Y@Lu/Dy@Y (red), Lu@Lu/Dy@Y NPs (cyan), and Lu@Lu/Dy@Lu (purple) NPs. **d** FFT pattern of a single Y@Lu/Dy@Y NP. **e** Strain maps of GPA ($\omega_{xy}$ and $\omega_{yy}$) for the (110) plane in the high-resolution TEM image. **f** Proposed energy transfer mechanism for the improved XEPL. Hot electrons and deep holes are generated through the photoelectric effect, with some secondary electrons being captured by traps, subsequently giving rise to XEPL emission. CB conduction band. Source data are provided as a Source Data file.

thereby reducing non-radiative relaxation processes, but also promotes the generation of anion Frenkel defects at the heterogeneous interfaces, which serve as additional traps upon X-ray irradiation. As a result, these core@shell@shell NPs with higher trap concentrations exhibit significantly stronger XEPL intensity.

## Universality study of the proposed mechanism

To verify the universality of the above proposed mechanism, the XEPL properties of additional lanthanide activators including Pr, Er, Tm, Gd, and Tb were studied. As depicted in Fig. 4a and Supplementary Figs. 25, 26, the XEPL intensities of all these lanthanide activators were greatly enhanced when employing core@shell@shell structures with dual interfaces, compared to the core NPs. Particularly, the XEPL intensity of Tb ions in the ultraviolet (UV) region was enhanced approximately 87.8 times (Fig. 4a). A visual confirmation was provided by the digital photographs (Fig. 4b), which clearly demonstrated stronger XEPL intensities and longer durations of these lanthanide activators in the core@shell@shell structures compared to the core NPs. For instance, the green XEPL emitted by the Y@Lu/Gd/Tb@Y NPs remained visible even after 60 min, whereas the Lu/Gd/Tb core NPs ceased emitting XEPL after only 5 min. Moreover, upon X-ray irradiation at 30 kV, the XEPL of the Y@Lu/Gd/Tb@Y NPs persisted for over 8 h (Supplementary Fig. 27). The XEPL decay curves further confirmed that the XEPL performance of the lanthanide activators could be significantly improved by constructing the core@shell@shell structures with dual interfaces (Fig. 4c–g). Under different X-ray irradiation conditions, the XEPL intensity of the Y@Lu/Gd/Tb@Y core@shell@shell NPs was stronger than that of commercial counterparts ($CaAl_2O_4$:Eu/Dy, $SrAl_2O_4$:Eu/Dy, and CaS:Eu), and the relative intensities were summarized in Supplementary Fig. 28. These findings suggest that the designed core@shell@shell structure has the potential application for the development of advanced X-ray-activated persistent luminescence NPs.

## Delayed XEPL-based 3D imaging

Enhancing XEPL efficiency is a crucial approach for reducing the input X-ray dose rate and minimizing radiation risks, which holds significant importance in scientific research and practical applications. The Y@Lu/Gd/Tb@Y NPs integrated into a poly-dimethylsiloxane (PDMS) flexible film via a simple solution-based technique demonstrated promising applications in low-dose and high-resolution 3D imaging. The flexible film exhibited excellent flexibility to be conformable, with the ability to bend up to approximately 180° and stretch more than 200% (Fig. 5a and Supplementary Fig. 29), while maintaining high optical transparency of over 90% in the range of 300–800 nm (Supplementary Fig. 30). To assess the imaging capabilities of the film, a standard resolution pattern plate was subjected to real-time X-ray imaging using a home-built X-ray imaging system (Supplementary Fig. 31). The imaging system achieved a high resolution of approximately 16.6 line pairs per millimeter (LP mm$^{-1}$), as confirmed by the analysis of gray values from light and dark stripes (Fig. 5b and Supplementary Fig. 32). Furthermore, the spatial resolution was determined to be approximately 17.1 LP mm$^{-1}$, with a modulation transfer function value of 0.2 (Fig. 5c). This spatial resolution surpassed that of most previously reported halide perovskite-based X-ray imaging systems[37] (typically lower than 10 LP mm$^{-1}$) and even outperformed a commercial CsI (Tl) scintillator[38] (≈10 LP mm$^{-1}$). Distinct from the traditional real-time X-ray imaging methods, the lanthanide-doped fluoride nanoscintillators with strong XEPL can be used for delayed 3D imaging. As schematically illustrated in Fig. 5d, a target object enclosed within a black box is initially wrapped with the prepared flexible film. The internal information of the target are visualized by capturing XEPL from many different directions. Finally, a 3D reconstruction procedure is employed to obtain the complete 3D structure of the target. In a practical demonstration, a real-time X-ray image of an

electronic watch exhibited several bright spots, potentially originating from the X-ray-excited optical luminescence of micro-LED chips (Fig. 5e and Supplementary Fig. 33). However, these spots were completely absent in the delayed X-ray image (Fig. 5f). Moreover, the internal electronic structures of the watch were clearly captured from five different directions (Fig. 5f and Supplementary Fig. 34). Consequently, the 3D reconstruction procedure yielded the internal 3D structure of the electronic watch with spatial resolution of approximately 25.64 LP mm$^{-1}$ (Fig. 5g and Supplementary Fig. 35). Notably, under the same X-ray irradiation dose rate of 4.5 μGy s$^{-1}$, the delayed X-ray image based on the Lu/Gd/Tb core and Lu@Lu/Gd/Tb@Lu NPs exhibited significantly lower clarity compared to the image obtained using Y@Lu/Gd/Tb@Y NPs (Supplementary Fig. 36). It is important to mention that in our case, we did not employ a heating procedure to accelerate the release of deposited X-ray energy.

## Discussion

In conclusion, our experimental study has demonstrated the amplification of XEPL intensities in lanthanide-doped fluoride NPs by constructing dual heterogeneous interfaces within a core@shell@shell nanostructure. This design strategy effectively passivates surface quenchers, reducing non-radiative relaxation, and enhances trap concentration by lowering the formation energy of interfacial Frenkel defects. By integrating the lanthanide-doped fluoride NPs into a flexible film, we have showcased their application as a conformable scintillation screen for delayed XEPL-based 3D imaging. We achieved clear visualization of the internal electrical structures of a watch through delayed X-ray imaging and subsequent 3D reconstruction. This highlights the potential of the studied lanthanide-doped fluoride NPs with bright XEPL for low-dose and high-resolution 3D imaging. These findings show promising application for the development of advanced X-ray-activated persistent fluoride NPs, with promising applications in delayed 3D imaging, biomedicine, displays, and information storage. Moreover, the enhanced XEPL intensities and improved imaging capabilities of these NPs offer opportunities for safer and more efficient X-ray imaging techniques in various scientific and practical fields.

## Methods

### Chemicals and reagents

Y(Ac)$_3$·xH$_2$O (99.9%), Gd(Ac)$_3$·xH$_2$O (99.9%), Lu(Ac)$_3$·xH$_2$O (99.9%), Dy(Ac)$_3$·xH$_2$O (99.9%), Tb(Ac)$_3$·xH$_2$O (99.9%), Pr(Ac)$_3$·xH$_2$O (99.9%), Er(Ac)$_3$·xH$_2$O (99.9%), Tm(Ac)$_3$·xH$_2$O (99.9%), NH$_4$F (98%), NaOH (≥96%), 1-octadecene (ODE, 90%), oleic acid (OA, 90%) and CHCl$_3$ (≥99.5%) were supplied by Sigma Aldrich Company. Cyclohexane, methanol and absolute ethanol were purchased from Sinopharm Chemical Reagent Company. Polydimethylsiloxane (PDMS, SYL-GARD™ 184 Silicone Elastomer Kit) was purchased from Dow Corning Company. The commercial $CaAl_2O_4$:Eu/Dy $SrAl_2O_4$:Eu/Dy, and CaS:Eu persistent phosphors were purchased from Xiucai Chemical and Andron Technologies company. All chemicals were of analytical grade and used as received without further purification.

### Synthesis of Lu/Gd/Dy core NPs

Lu(Ac)$_3$ (0.624 mmol), Gd(Ac)$_3$ (0.16 mmol) and Dy(Ac)$_3$ (0.016 mmol) were added to a 50 mL three-necked round-bottom flask containing 8 mL OA. The mixture was heated at 145 °C for 30 min to remove any water from the solution. Next, 12 mL ODE was quickly injected into the above solution which was then heated at 150 °C for 60 min to form a transparent solution, followed by cooling down to room temperature. Afterward, 8 mL MeOH solution containing NH$_4$F (3 mmol) and NaOH (2, 8 mmol) was added dropwise to the cooled solution while stirring at 60 °C for 40 min. After the MeOH was evaporated completely, the solution was heated to 300 °C under N$_2$ atmosphere and kept for 90 min before being cooled down to room temperature. The products were precipitated by adding EtOH, collected by centrifugation, washed

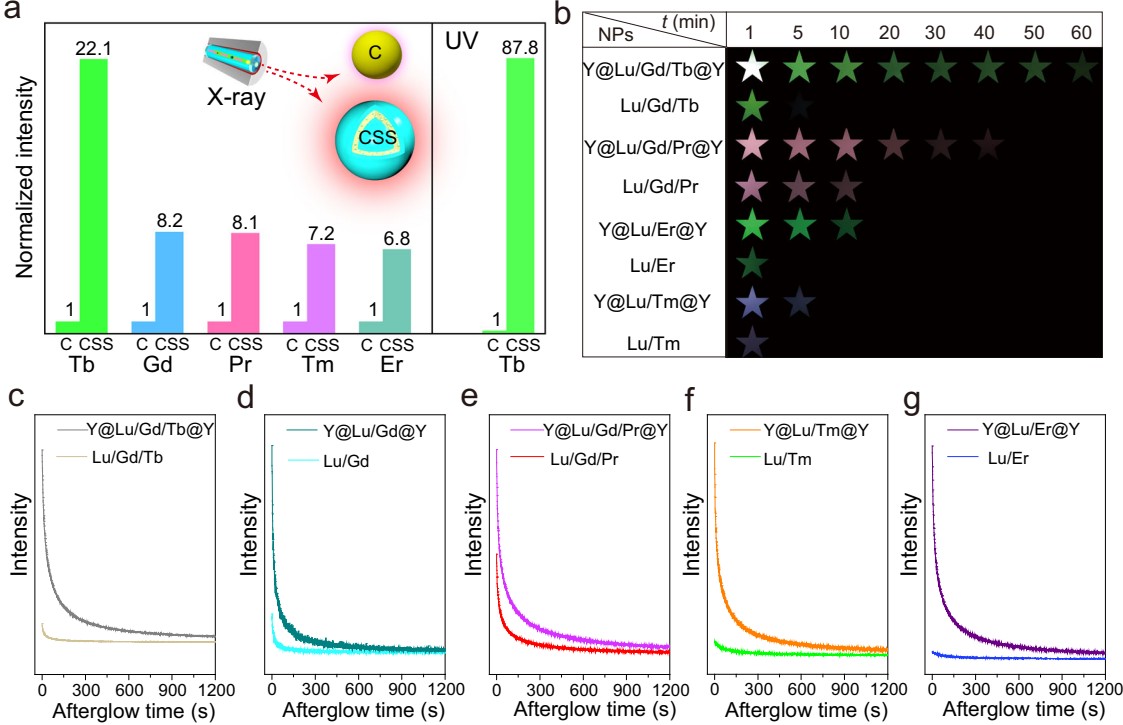

**Fig. 4 | XEPL characterizations of more lanthanide activators doped in the heterogeneous core@shell@shell structure. a** Integral XEPL intensities of different lanthanide activators-doped core ([Na]/[RE] = 2.5) and heterogeneous core@shell@shell NPs. The XEPL intensities for the core NPs were normalized to 1. C, core; CSS, core@shell@shell. Inset, XEPL intensity of core NP is much weaker than that of CSS NP. **b** Photographs of those core and core@shell@shell NPs at different times after the cessation of X-rays. XEPL decay curves of those core and core@shell@shell NPs doped with different lanthanide activators, Tb (**c**), Gd (**d**), Pr (**e**), Tm (**f**), Er (**g**). X-ray operation was set at a voltage of 30 kV. Source data are provided as a Source Data file.

with EtOH and cyclohexane for three times, and finally dried or re-dispersed in 4 mL cyclohexane for further use.

### Synthesis of NaYF₄ core NPs
Y(Ac)₃ (0.8 mmol) were added to a 50 mL three-necked round-bottom flask containing 8 mL OA. The mixture was heated at 145 °C for 30 min to remove water from the solution. Then 12 mL ODE was quickly added into the above solution which was then heated at 150 °C for 60 min to form a transparent solution, and then cooled down to room temperature. Afterward, 8 mL MeOH solution containing NH₄F (3 mmol) and NaOH (2 mmol) was added dropwise to the cooled solution while stirring at 60 °C for 40 min. After the MeOH was fully evaporated, the solution was heated to 290 °C under N₂ atmosphere and kept for 60 min, and then cooled down to room temperature. The products were precipitated by adding EtOH, collected by centrifugation, washed with EtOH and cyclohexane for three times, and finally re-dispersed in 4 mL cyclohexane for further use.

### Synthesis of Y@Lu/Gd/Dy core@shell NPs
Lu(Ac)₃ (0.624 mmol), Gd(Ac)₃ (0.16 mmol) and Dy(Ac)₃ (0.016 mmol) were added into a 50 mL three-necked round-bottom flask containing OA (8 mL). The mixture was heated at 145 °C for 30 min to remove water from the solution. Next, a solution of ODE (12 mL) was quickly added and the resulted mixture was heated at 150 °C for 60 min to form a clear solution, and finally cooled down to 80 °C. The pre-prepared NaYF₄ core NPs in 4 mL cyclohexane was injected to the above solution and kept at 100 °C for 40 min. After the removal of cyclohexane, followed by cooling down to room temperature, 8 mL MeOH solution containing NH₄F (3 mmol) and NaOH (8 mmol) was added dropwise to the solution while stirring at 60 °C for 40 min. After the MeOH was fully evaporated, the solution was heated at 300 °C under N₂ for 90 min before being cooled down to room temperature.

The products were precipitated by adding EtOH, collected by centrifugation, washed with EtOH and cyclohexane for three times, and finally re-dispersed in 4 mL cyclohexane.

### Synthesis of Y@Lu/Gd/Dy@Y core@shell@shell NPs
Y(Ac)₃ (0.8 mmol) was added into a 50 mL three-necked round-bottom flask containing OA (8 mL). The mixture was heated at 145 °C for 30 min to remove water from the solution. Then, a solution of ODE (12 mL) was quickly added and the mixture was heated at 150 °C for 60 min to form a clear solution, which was then cooled down to 80 °C. The pre-prepared NaYF₄@NaLuF₄:20Gd/2Dy core-shell NPs in 4 mL cyclohexane was injected into the above solution and kept at 100 °C for 40 min. After the removal of cyclohexane, 8 mL MeOH solution containing NH₄F (3 mmol) and NaOH (2 mmol) was added dropwise to the solution while stirring at 60 °C for 40 min. After the MeOH was evaporated completely, the solution was further heated at 300 °C under N₂ for 90 min before being cooled down to room temperature. The products were precipitated by adding EtOH, collected by centrifugation, washed with EtOH and cyclohexane for three times, and finally dried at 60 °C.

### Preparation of PDMS film with scintillators
The 150 mg NPs were dispersed in CHCl₃ by stirring for 20 min at room temperature. Then, 1.1 g PDMS (prepolymer and crosslinker in a 10:1 weight ratio) was mixed with the above solution and sonicated for 40 min to ensure a uniform distribution of the NPs. Afterward, the solution containing NPs was spin-coated onto a cover slide substrate at 6000 rpm and solidified at 70 °C for 12 h.

### Delayed XEPL-based 3D imaging
The object was enveloped in a flexible scintillator film and exposed to X-rays. Once X-ray exposure ceased, we captured images from

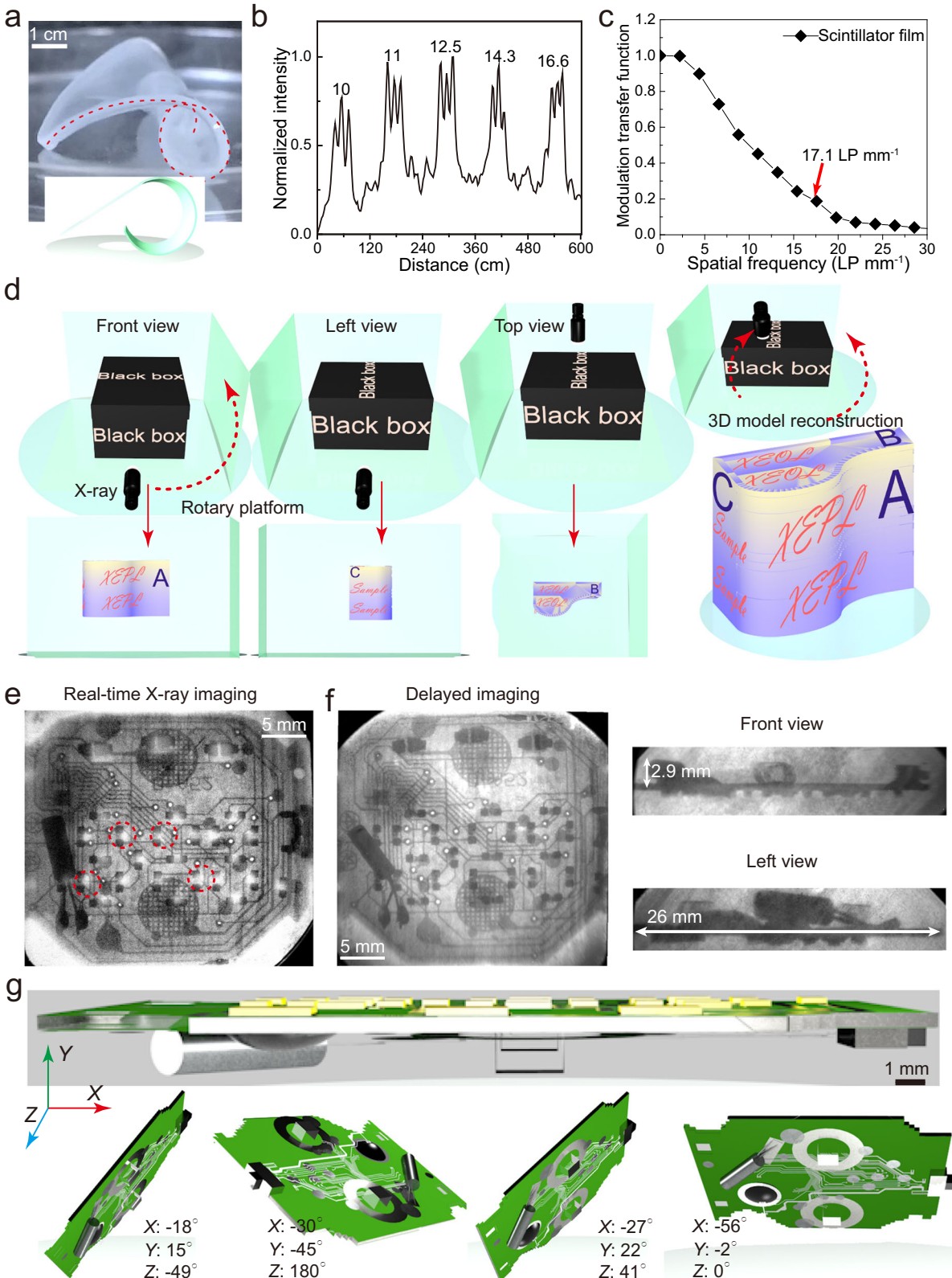

**Fig. 5 | Delayed XEPL-based 3D imaging. a** photograph of the as-prepared flexible film. **b** The grayscale function of real-time X-ray image ranging from 10.0 to 16.6 LP mm⁻¹. **c** MTF curve of the real-time X-ray image obtained from the flexible film. LP, line pairs. **d** Schematic showing delayed XEPL-based 3D imaging enabled by a flexible film. The internal information of the target is visualized by capturing XEPL from different directions. Then, a 3D reconstruction procedure is employed to obtain a complete 3D structure of the target. **e** Real-time X-ray image of a digital watch. **f** Delayed XEPL-based images from different directions. **g** 3D photograph of the electronic watch via a 3D reconstruction procedure. Source data are provided as a Source Data file.

different angles using a digital camera (ORCA-Fusion BT, pixel size: $6.5\,\mu m \times 6.5\,\mu m$). The 3D model structures were then reconstructed using Maxon Cinema 4D (R26).

## Characterizations

XRD analysis was carried out by a powder diffractometer (Bruker D8 Advance) with a Cu-Kα ($\lambda = 1.5405\,\text{Å}$) radiation. The morphology and size of the products were characterized by a field-emission transmission electron microscopy (TEM, FEI Tecnai G2 F20) equipped with an energy dispersive X-ray spectroscopy (EDS, Aztec X-Max 80 T). Scanning transmission electron microscopy (STEM) imaging was performed on a spherical aberration-corrected STEM instrument, ThermoFisher Spectra 300, equipped with a 5th order aberration corrector. The beam current of the electron probe is set to be ≈50 pA and a typical dose level of ≈$0.4-1 \times 10^8$ e nm$^{-2}$ as used for high-resolution imaging. Afterglow spectra were recorded on an OmniFluo-Xray-JL system (PMT-CR131-TE detector, 185-900 nm) with a mini MAGPRO X-ray excitation source. X-ray photoelectron spectroscopy analyses were performed using Thermo Fisher Scientific K-Alpha with tube voltage 15 kV and tube current 15 mA. To measure the TL glow curves, the NPs were exposed to the X-ray source for 60 s at room temperature. Afterwards signals were recorded by using the LTTL-3DS multifunctional defect fluorescence spectrometer with tube voltage of 50 kV and tube current of 80 µA, while the temperature range was set between 30 and 500 °C at a heating rate of 2 °C s$^{-1}$. EPR was carried out using a Bruker model A300 spectrometer recorded at 9.852 GHz.

## Computation method

All the calculations were performed in the framework of the DFT with the projector augmented plane-wave method, as implemented in the Vienna ab initio simulation package[39]. The generalized gradient approximation was selected for the exchange-correlation potential[40]. The cut-off energy for plane wave was set to 420 eV. The energy criterion was set to $10^{-5}$ eV in iterative solution of the Kohn-Sham equation. A $1 \times 1 \times 2$ supercell of NaLuF$_4$ contains 3 Na atoms, 3 Lu atoms and 12 F atoms, and A $1 \times 1 \times 2$ supercell of NaYF$_4$ contains 3 Na atoms, 3 Y atoms and 12 F atoms were employed. As for the slab models, the (100) plane was selected for both NaLuF$_4$ and NaYF$_4$ to construct heterojunction structures of NaLuF$_4$@NaYF$_4$, NaLuF$_4$@NaLuF$_4$ and NaYF$_4$@NaYF$_4$. The vacuum space along the $z$-direction was set to be over 15 Å, which was enough to avoid interaction between the two neighboring images. The Gamma k-point mesh was sampled with a separation of about 0.03 Å$^{-1}$ in the Brillouin zone. All the structures were relaxed until the residual forces on the atoms had declined to less than 0.05 eV Å$^{-1}$.

## Reporting summary

Further information on research design is available in the Nature Portfolio Reporting Summary linked to this article.

# Data availability

The data that support the findings of this study are available from the corresponding authors upon request. Source data are provided with this paper.

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

## Acknowledgements

This work is supported by National Natural Science Foundation of China (No. 52172164), Fundamental Research Funds for the Provincial Universities of Zhejiang (2023YW72). The Institute for Lasers, Photonics and Biophotonics acknowledges support from the office of Vice President for Research and Economic Development at the University at Buffalo.

## Author contributions

L.L. and M.Y. initiated and designed the project. M.Y. performed nanoparticle synthesis. M.Y. and Y.W. conducted optical measurements and taken photographs. L.L., P.P and S.X. wrote and revised the manuscript. L.L., M.Y., Y.W., Y.H., J.Z., P.P. and S.X. contributed to the data analyses and discussion.

## Competing interests

The authors declare no competing interests.
