## [Peer Review File · Nature Communications]

Dual heterogeneous interfaces enhance X-ray excited persistent luminescence for low-dose 3D imagingReviewers' comments:

Reviewer #1 (Remarks to the Author):

This paper has cast light upon X-ray scintillating lanthanide doped fluoride nanoparticles (NPs), which showed related applications in three dimensional (3D) imaging by fabricating flexible X-ray detectors. The X-ray excited persistent luminescence can get improved for the construction of dual heterogeneous interfaces in the core/shell/shell nanostructure. However, we do not consider it suitable for the publication on nature communications. Reasons are listed as follows:

1. In the Delayed XEPL-based 3D imaging, the authors used a home-built X-ray imaging system for real-time imaging. It is quite amazing that the 3D model reconstruction could be achieved from merely three different directions. For 3D X-ray imaging, the imaged object should be placed on different rotating stage for highly-resolution, see ACS Applied Materials & Interfaces 2023 15 (1), 932-941 for more details. Or like the cine-CT, the X-ray tube, objects and detectors (mainly scintillators) are placed in a line and the X-ray tube are desired to move around the imaged objects to collect many radio-photos for high-resolution 3D model reconstruction.
2. For real-time X-ray imaging, especially for 3D imaging. The scintillation afterglow or the decay time should be shorter, which indicates that the fast scintillators with high luminance are better for dynamic real-time imaging, see PhotoniX, 2022, 3, 9 for details. How can the Y_{2.5}@Tb₁₀@Y_{2.5} NPs, the super long persistent materials with delayed 8 hour luminescence achieve dynamic real-time imaging without overlap and/or halation and/or blurry figure?
3. The important data of light yields and the detection limits of as-prepared related NPs were missing, see recently published literatures for details, like ACS Cent. Sci. 2023, DOI: 10.1021/acscentsci.3c00563 and ref 9.
4. As the literature Nat. Mater., 2023, 22, 289–304 pointed out, the post-annealing treatment, surface passivation, dyes sensitization and surface plasmon resonance can all enhance the persistent luminescence in in nanocrystalline phosphors. It is a usual strategy for the core/shell/shell nanostructure benefited from surface passivation to gets the enhanced luminescence.
5. It seems that the persistent luminescence can be rooted in the spectra multiplexing, not the interfacial passivation, for different rare earth ions were well isolated in different layers in the core/shell/shell nanostructure. The authors did not tell details on how to distinguish the persistent luminescence mechanism between them two or other possible ones.
6. Could the author give direct evidence that the incompletely coordinated surface F⁻ ions facilitate the formation of Frenkel defects upon ionizing radiation. In the nanoparticles, why and how the F⁻ ions could facilitate the Frenkel defects, as the LuF₄⁻ seems to form the Frenkel excitons (J. Phys. Chem. Lett. 2013, 4, 11, 1788–1796). Upon radiation of X-rays, the heavy rare earth ions in the outer-shell are more vulnerable to be ionized to generate hot electrons for their higher X-ray absorption cross section. And the photogenerated electrons could further bring forth electron-hole pairs, which are bound by Coulomb force, leading to the formation of Frenkel excitons.

Reviewer #2 (Remarks to the Author):

The paper is focused on the behaviour of X-ray excited persistent luminescence (XEPL) of lanthanide activated core and multishell fluoride nanoparticles (NPs). The XEPL results enhanced by careful varying the architecture of the nanoparticles, type of dopant and its concentration. The paper also describes an interesting investigation of the XEPL enhancement, mainly of Dy³⁺ doped NPs, due to heterogeneous interfaces among core and shell layers, considering the generation of traps upon X-ray irradiation. An investigation concerning the XEPL for other lanthanide ions is also reported, evidencing a strong enhancement of the XEPL intensity for the core/shell/shell structure with respect to the core one. The high resolution 3D imaging investigation using a polymer based flexible film loaded with the NPs is also interesting for the scientific community. The paper deserves to be published in Nature

Communication. I have only a couple of minor issues:

- the authors state that the dual heterogeneous interface promotes the generation of anion Frenkel defects at the heterogeneous interface: have the authors other experimental evidences of the increase of these defects ? Is it possible at least to evaluate their concentrations in the core and core/shell (/shell) structures ?

- what is the spatial resolution that could be achieved in the 3D reconstruction ? This is an interesting value to know in order to exploit this system in real technological applications.

As a conclusion, the paper can be accepted after minor revision.

Reviewer #3 (Remarks to the Author):

The investigation of nanoparticles that show tunably x-ray excited persistent luminescence (XEPL) is certainly a very active and promising topic of applied nanoparticle research. The authors show here very interesting results by which they are able to enhance XEPL intensity signals largely by building nanoparticles with a well-designed core-shell-shell structure. In addition, the authors also give good experiments to gain an insight into the mechanism which is behind the large intensity enhancement of their core-shell-shell nanoparticles. In general, it can be stated that the manuscript is well structured and written very clearly. The characterisation done on the nanoparticles, as well as their application in sensing are done in high quality and the results are very convincing. Also the methods employed and how the results were obtained is described in a fully clear way and is well meeting high standards in the field.

Accordingly, in general the claims of this work are well substantiated. However, in order to appreciate further the enhancement of XEPL intensity reported here, it would be interesting to not just compare the outcome relative within the series of nanoparticles produced by the authors. As the intensity is given in a.u. it is impossible to really appreciate the quality of the findings of this work, as one cannot compare the outcome to other work in the field. However, this is of central importance. Accordingly, one would either have to quantify the XEPL intensity in absolute terms compared to the x-ray intensity input or to compare under identical conditions with the performance of other, already published, nanoparticles (systems).

In summary, I would favour the publication of this contribution, but only once the claimed luminescence enhancement becomes clarified in absolute terms, so that it can really become compared to other comparable systems.

Reviewer #4 (Remarks to the Author):

NCOMMS-23-30665

Review report

In their manuscript (NCOMMS-23-30665), Lei and co-workers report on the formation of heterogeneous interfaces in core-shell β -NaREF₄ (where RE = rare earth elements) to improve X-ray excited persistent luminescence (XEPL). The authors also describe an example of delayed XEPL-based 3D imaging.

1. General comments and recommendation

1.1. Key results

The authors claim that the formation of heterogeneous dual interfaces in core-shell NaREF₄ materials enhances X-ray excited persistent luminescence. The authors managed to perform delayed X-ray imaging of the internal structure of a watch.

1.2. Validity of the data and conclusions

The quality of the experimental data is relatively poor and important data (x-ray diffraction, STEM

images, chemical maps, EDX line scan) are missing for a large number of samples that are described in the manuscript. This prevents to properly compare the different samples. Favoring classical TEM instead of STEM is a methodological mistake especially when claiming differences between homogeneous and heterogeneous interfaces in core-shell materials. Finally, most of authors' claims are not supported by experimental evidence (see section 4) and most recent discoveries regarding core-shell NaREF₄ materials (see section 3) are not considered by the authors leading to misinterpretations (see section 4).

1.3. Significance

The submitted manuscript appears to be an extension of authors' previous work (where XEPL was called XEA – X-ray excited afterglow), which was published in 2022 (Nature Communications 2022, 13, 5739). The heterogeneous aspect, supposed to be the main novelty of the submitted manuscript, was already described (to a certain extent) in Nature Communications (2022, 13, 5739). This is also the case for the mechanistic investigation (see details below), which is exactly the same as reported in Nature Communications (2022, 13, 5739). This combined to the fact that the conclusions are not supported by the experimental data, the significance of the manuscript is very limited.

1.4. Recommendation

The submitted manuscript should be rejected.

2. Clarity and context

2.1. The manuscript is poorly written and extremely hard to understand even for someone in the field. The choice of samples' abbreviations makes the reading very unpleasant and the reader constantly has to refer to Table 1.

2.2. Figures' captions, are not self-explanatory and the reader must dig into the text to understand the figures. It is absolutely impossible to understand the figures with the chosen abbreviations.

2.3. Authors botched supplementary figures' captions.

2.4. In the main text and figures' captions, authors alternate between abbreviations and full names. Authors should stick to only one type throughout the text.

2.5. There is no proper comparison with already published articles focused on X-ray excited luminescence imaging.

3. References

3.1. The manuscript does not appropriately refer to the existing literature. In particular, authors completely ignored intermixing problems observed with NaREF₄ core-shell materials (e.g. Chem. Mater. 2015, 27, 8375; Small 2021, 17, 2104441; J. Am. Chem. Soc. 2023, <https://doi.org/10.1021/jacs.3c03019>).

4. Scientific aspects

4.1. "The XEPL intensity of the Y_{2.5}@Dy₁₀@Y_{2.5} NPs was approximately 40.9 times higher than that of Dy_{2.5} NPs, where the subscript numbers represent their fractions in the composition."

Comparing core-shell nanoparticles with pure core nanoparticles is not a fair comparison due to significantly enhanced surface quenching in the latter case. What did you expect?

4.2. "The XEPL intensity of the Y_{2.5}@Dy₁₀@Y_{2.5} NPs was approximately 40.9 times higher than that of Dy_{2.5} NPs, where the subscript numbers represent their fractions in the composition."

The underlined text is absolutely not clear and does not mean anything. It also seems to be wrong because when reading the text it seems that 2.5 and 10 refer to the [Na]/[RE] ratio.

4.3. "All these NPs were well indexed to a pure hexagonal phase without any impurity peaks (Fig.

S2)“

How do the authors explain that Bragg peaks of 35 nm core nanoparticles (Dy_{2.5}) are as sharp (or very likely even sharper) as Bragg peaks of larger (62-65 nm) core-shell nanoparticles (Dy₁₀@Lu_{2.5}, Dy₁₀@Y_{2.5}, Y_{2.5}@Dy₁₀@Y_{2.5})?

4.4. “The Y element was observed in the Dy₁₀@Y_{2.5} core/shell NPs (Fig. S3d), but not in the Dy₁₀ core layer (Fig. S3b), indicating the formation of a core/shell structure.”

A basic large area EDX analysis performed on a large number of particles (supplementary figure 3) does not prove the formation of core-shell structures. Why would you like to find Y in pure core nanoparticles (Dy₁₀) that do not contain Y (NaLuF₄:20Gd/2Dy according to Table 1)? This sentence makes no sense.

4.5. “The core/shell/shell structure of the Y_{2.5}@Dy₁₀@Y_{2.5} NPs was clearly verified by the high angle annular dark-field (HAADF) TEM image, which is sensitive to the atomic numbers^{29,30} (Fig. 1b)”

Scanning transmission electron microscopy is not just sensitive to the chemical composition but also to the thickness. The STEM image presented in figure 1b is given with false colors. The reviewer wants to see the corresponding raw STEM image. Performing HAADF-STEM is important and must be performed on ALL samples reported in the manuscript including low magnification to assess the homogeneity of the synthesized nanoparticles. This is not the case and authors mostly included TEM images, which are not as informative as STEM images when dealing with core-shell nanoparticles.

4.6. “EDX line scan and mapping analysis revealed that Lu was primarily distributed in the interlayer and Y was located in the core and the outer shell (Fig. 1c), which were consistent with the nominal compositions of the core/shell/shell structure.”

The reviewer would like to see the STEM image from which the chemical maps presented in figure 1c have been obtained. The STEM image must be given at least in the supplementary information. The reviewer would like to see the Y and Gd maps (individual channels), which should be added in the supplementary information. Finally, the Lu chemical map makes no sense for a core-shell-shell nanoparticle. Indeed, Y_{2.5}@Dy₁₀@Y_{2.5} refers to NaYF₄@NaLuF₄:Gd:Dy@NaYF₄. This means that Lu (and Gd) are confined (according to the authors) in the interlayer. Therefore, it is impossible to get nearly no Lu counts in the central region of the particles as demonstrated with completely black areas. Can the authors clarify? Did the authors perform post-processing to clean up images? The line scan analysis is also not in agreement with the formation of core-shell-shell nanoparticles because the Y signal has a very similar magnitude compared to the Lu signal on the sides of the particle, which makes no sense because the external shell is supposed to be a pure Y shell. This can be a sign of cation intermixing, which was not considered by the authors.

4.7. “Increasing the [Na]/[RE] ratio in the precursor solution from 2.5 to 10 resulted in a 6.8-fold increase in the XEPL intensity (Fig. 1d-e)”

The authors did not quantify the [Na]/[RE] ratio. This should be done for all reported samples in the manuscript.

4.8. “Interestingly, the XEPL intensity of the Dy₁₀@Y_{2.5} NPs was much stronger than that of the Dy₁₀@Lu_{2.5} NPs with similar core sizes and shell thicknesses, indicating that the heterogeneous interface produced an improvement in the XEPL intensity.”

This is pure speculation. The authors only presented basic TEM images (supplementary figure 1) instead of STEM images. Chemical maps and EDX line scans should be performed to validate the core-shell formation. The Gd signal can be utilized (due to its relatively high concentration of 20%) even for the homogeneous nanoparticles.

4.9. “Confirming the enhancement of XEPL by designing comparable nanostructures”

For this second set of samples, the authors added Dy in the interlayer and not in the core compared to the first set of samples for which Dy was in the core. Moreover, Gd has been added in the first set of samples but not in the second. Therefore, authors do not compare absolutely equivalent homogeneous

(set #2) and heterogeneous (set #1) nanoparticles. It is also obvious from figure 2 that the quality of the synthesized homogeneous (figures 2a-b) and heterogeneous (figures 2d-f) nanoparticles is not equivalent. See for instance figures 2a(bottom) and 2e or figures 2b and 2f. Equally important, all corresponding STEM images, chemical maps, and EDX line scans are missing for samples presented in Figure 2. Those are of major interest especially when dealing with core-shell structures.

4.10. The mechanistic investigation described in the manuscript is exactly the same as the one described in Nature Communications 2022, 13, 5739. Are there differences?

More importantly, the authors never considered cation intermixing and the modification of real concentrations including non-homogeneous distributions and concentration gradients (Small 2021, 17, 2104441; J. Am. Chem. Soc. 2023, <https://doi.org/10.1021/jacs.3c03019>) despite the fact that their experimental data do not allow to exclude such a phenomenon.

4.11. Figure 3d: the corresponding indexation of the FFT is missing.

4.12. "To confirm the presence of lattice distortion in the heterogeneous 196 interface, a fast Fourier transform (FFT) pattern analysis was conducted. The FFT pattern revealed two distinct sets of diffraction spots corresponding to the core and the interlayer of a single NP, indicating a difference in lattice constants between the two regions (Fig. 3d). This observation suggests the presence of lattice distortion at the interface³³. A geometric phase analysis (GPA) of the high-resolution TEM image of a single core/shell/shell NP was employed to visualize the strain distribution^{22,34}."

Pure speculation. You do not have the necessary resolution for such measurements (just look at the FFT presented in figure 3d). Let's assume authors managed to distinguish 2 sets of diffraction spots, this means authors should have seen 2 sets of Bragg peaks by X-ray diffraction. This is not the case because of the overlap due to the small size of the coherent domains. Broad peaks in X-ray diffraction also means diffuse spots in electron diffraction. What are the calculated values of the extracted cell parameters and what are the corresponding errors and standard deviations?

4.13. What are the results when performing delayed XEPL-based 3D imaging with homogeneous core-shell nanoparticles that are strictly equivalent to their heterogeneous counterparts?

Response to reviewer's comments

We greatly appreciate the reviewers' insightful comments which were very helpful for the improvement of our manuscript. In response to the valuable comments raised by the referees, we provide point-by-point responses along with the modifications (marked in blue) made in the revised manuscript.

Reviewer 1:

Comment: In the Delayed XEPL-based 3D imaging, the authors used a home-built X-ray imaging system for real-time imaging. It is quite amazing that the 3D model reconstruction could be achieved from merely three different directions. For 3D X-ray imaging, the imaged object should be placed on different rotating stage for highly-resolution, see ACS Applied Materials & Interfaces 2023 15 (1), 932-941 for more details. Or like the cine-CT, the X-ray tube, objects and detectors (mainly scintillators) are placed in a line and the X-ray tube are desired to move around the imaged objects to collect many radio-photos for high-resolution 3D model reconstruction.

Response: We are sincerely grateful to the reviewer for this valuable comment, which has significantly contributed to the improvement of our work. Following a thorough examination of the suggested reference (ACS Applied Materials & Interfaces 2023 15 (1), 932-941), we have come to realize that there may be multiple potential outcomes for 3D model reconstruction based on images acquired from only three angles when dealing with complex objects. The contrast in X-ray images can provide critical depth information, thereby greatly reducing the number of possible reconstruction scenarios, particularly for simpler devices. In our study, we have employed a movable X-ray tube that can be positioned around the imaged object to capture delayed images from multiple directions, similar to cine-CT. To ensure the accuracy and reliability of our 3D model reconstruction, we have collected XEPL-based delayed images from more different angles.

The Fig. 5d has been revised and additional delayed XEPL-based images (Fig. S33 and S34) have been added in the revised manuscript and supporting information.

Fig. 5 Delayed XEPL-based 3D imaging. **g** Delayed XEPL-based images from different directions.

Supplementary Fig. 35 Delayed XEPL-based images from different directions.

Supplementary Fig. 36 Reconstructed models based on the delayed images from different directions.

Comment: For real-time X-ray imaging, especially for 3D imaging. The scintillation afterglow or the decay time should be shorter, which indicates that the fast scintillators with high luminance are better for dynamic real-time imaging, see PhotoniX, 2022, 3, 9 for details. How can the $Y_{2.5}@Tb_{10}@Y_{2.5}$ NPs, the super long persistent materials with delayed 8 hour luminescence achieve dynamic real-time imaging without overlap and/or halation and/or blurry figure?

Response: We appreciate the reviewer's emphasis on the importance of fast scintillators with high luminance for dynamic real-time X-ray imaging. We

acknowledge that traditional real-time imaging benefits immensely from fast scintillators. However, it is essential to emphasize that our study is dedicated to highlighting the unique advantages of lanthanide-doped fluoride nanoscintillators, which exhibit robust XEPL. While they may not be suitable for conventional dynamic real-time X-ray imaging, they excel in delayed XEPL-based 3D imaging. The key factor for high-quality delayed images is the exceptional XEPL performance exhibited by these nanoscintillators, which sets them apart from fast scintillators commonly used in real-time imaging scenarios.

To provide a clearer understanding of the delayed XEPL-based 3D imaging processes, we have included the following details into the revised manuscript:

"Delayed XEPL-based 3D imaging

The object was enveloped in a flexible scintillator film and exposed to X-rays. Once X-ray exposure ceased, we captured images from different angles using a digital camera (ORCA-Fusion BT, Pixel size: $6.5 \mu\text{m} \times 6.5 \mu\text{m}$). The 3D model structures were then reconstructed using the Cinema 4D software."

Comment: The important data of light yields and the detection limits of as-prepared related NPs were missing, see recently published literatures for details, like ACS Cent. Sci. 2023, DOI: 10.1021/acscentsci.3c00563 and ref 9.

Response: Light yield and detection limit are crucial parameters in the field of dynamic real-time X-ray imaging. For delayed X-ray excited persistent luminescence (XEPL)-based 3D imaging, brightness and long persistent luminescence are equally significant. In light of the valuable suggestion from the reviewer, we further investigated the light yield and detection limit of the Y@Lu/Gd/Tb@Y NPs. Due to the relatively long lifetime of lanthanide activators (up to milliseconds), accurately measuring the light yield of lanthanide-doped fluoride NPs through conventional pulse height measurement methods presents challenges. To address this, we evaluated the relative scintillation intensity of the Y@Lu/Gd/Tb@Y NPs, using commercial CsI (TI) and GOS:Tb scintillators as references. As illustrated in the Figure R1, the scintillation intensity of the Y@Lu/Gd/Tb@Y NPs was approximately 1.06 times that

of GOS:Tb and 0.66 times that of CsI (Tl). The detection limit was measured to be approximately 15.62 nGy s^{-1} (Figure R2).

Figure R1 SEM images of the commercial CsI:Tl (a) and GOS:Tb (b) scintillators. (c) X-ray excited scintillation spectra of the Y@Lu/Gd/Tb@Y, commercial CsI:Tl and GOS:Tb scintillators. EDX spectra of commercial CsI:Tl (d) and GOS:Tb (e) scintillators. (f) Compared integral scintillation intensity ratios of Y@Lu/Gd/Tb@Y to CsI:Tl and Y@Lu/Gd/Tb@Y to GOS:Tb.

Figure R2 Scintillation intensity of the Y@Lu/Gd/Tb@Y core/shell@shell NPs

measured at low-dose rates. The detection limit is derived from the fitting curve when the SNR equals 3.

Considering this work focuses on the improvement of XEPL intensity of the lanthanide doped fluoride NPs and the application of XEPL-based delayed imaging, we further compared the XEPL intensity of the Y@Lu/Gd/Tb@Y core@shell@shell NPs with those of commercial persistent phosphors, namely, CaAl₂O₄:Eu/Dy, SrAl₂O₄:Eu/Dy, and CaS:Eu, under various X-ray irradiation conditions. The comparative XEPL intensities among these persistent phosphors are summarized in **Supplementary Fig S28**. With an increase in irradiation energy, the generation of secondary electrons also increases, resulting in enhanced XEPL. Due to differences in the variation trends among these systems, the relative XEPL intensities varied under different irradiation energies.

The following contents have been added in the revised manuscript.

The commercial CaAl₂O₄:Eu/Dy SrAl₂O₄:Eu/Dy, and CaS:Eu persistent phosphors were purchased from Xiucui Chemical and Andron Technologies company.

Under different X-ray irradiation conditions, the XEPL intensity of the Y@Lu/Gd/Tb@Y core@shell@shell NPs was stronger than that of commercial counterparts (CaAl₂O₄:Eu/Dy, SrAl₂O₄:Eu/Dy, and CaS:Eu), and the relative intensities were summarized in Fig. S28.

Supplementary Fig. 28 Scanning electron microscope images of commercial persistent phosphors, CaS:Eu (a), CaAl₂O₄:Eu/Dy (b), SrAl₂O₄:Eu/Dy (c). EDX spectra of the CaS:Eu (d), CaAl₂O₄:Eu/Dy (e), SrAl₂O₄:Eu/Dy (f). XEPL spectra under different irradiation conditions, 20 KV (g), 30 KV (h), 40 KV (i), 50 KV (j). k Compared XEPL intensity ratios under different irradiation conditions.

Comment: As the literature Nat. Mater., 2023, 22, 289–304 pointed out, the post-annealing treatment, surface passivation, dyes sensitization and surface plasmon resonance can all enhance the persistent luminescence in in nanocrystalline phosphors. It is a usual strategy for the core/shell/shell nanostructure benefited from surface passivation to gets the enhanced luminescence.

Response: We appreciate the reviewer's insightful comment and for providing a relevant reference to this work. It is indeed true that there exist various strategies to enhance the persistent luminescence of nanocrystalline phosphors, such as post-annealing treatment, surface passivation, dyes sensitization, and leveraging surface plasmon resonance. While a typical core/shell structure can reduce energy migration efficiency from activators to surface quenchers and inhibit the formation of Frenkel defects by restoring the surface F⁻ ions coordination environment, it is challenging to significantly enhance XEPL intensity through the coating of a conventional shell layer. In our study, we ventured beyond conventional approaches by implementing a dual heterogeneous interface in our core@shell@shell structures. This innovative design served two pivotal purposes: firstly, it mitigated non-radiative relaxation possibilities by effectively passivating surface quenchers. Secondly, it reduced the interfacial Frenkel defect formation energy, leading to an increased trap concentration due to the presence of misfit strain at the heterogeneous interfaces. Additionally, we observed that the introduction of interstitial Na⁺ ions within the interfacial layer further enhanced XEPL intensity.

In summary, our core@shell@shell structures represent a novel and distinct approach in the field of XEPL, moving beyond the conventional core@shell structures typically reported in previous literatures. We believe that our innovative approach has the potential to contribute significantly to the development of XEPL materials and their applications.

Comment: It seems that the persistent luminescence can be rooted in the spectra multiplexing, not the interfacial passivation, for different rare earth ions were well isolated in different layers in the core/shell/shell nanostructure. The authors did not

tell details on how to distinguish the persistent luminescence mechanism between them two or other possible ones.

Response: The fundamental principle underlying persistent luminescence is the release of the trapped electrons (Nature 590, 410-415 (2021)). It is not a multiplexed emission comprised of emissions with different lifetimes originating from different centers. The XEPL in lanthanide doped fluoride NPs is generated through the process of detrapping of carriers and subsequent recombination, which is a relatively longer time duration process. This detrapping process is highly temperature-dependent. As depicted in Fig. 3c, after X-ray exposure ceases, the persistent luminescence intensity of the core@shell@shell NPs increases with rising temperature from room temperature to 428 K. This increase is due to the rapid release of abundant trapped electrons. Furthermore, XEPL spectra also reveal an increase in persistent luminescence intensity as the temperature rises from room temperature to 60 °C (Figure R3).

Consequently, persistent luminescence is not attributed to spectral multiplexing. In our study, we have primarily focused on enhancing the generation of anion Frenkel defects within lanthanide-doped fluoride nanocrystal structures. The XEPL enhancement mechanism resulting from these structural modifications has been thoroughly investigated and explained in detail in the main text.

Fig. 3c X-ray activated thermal luminescence spectra of the Y@Lu/Dy@Y,

Lu@Lu/Dy@Y NPs, and Lu@Lu/Dy@Lu NPs.

Figure R3 XEPL spectra of the Y@Lu/Dy@Y core@shell@shell NPs at room temperature and 60 °C.

Comment: Could the author give direct evidence that the incompletely coordinated surface F⁻ ions facilitate the formation of Frenkel defects upon ionizing radiation. In the NPs, why and how the F⁻ ions could facilitate the Frenkel defects, as the LuF₄⁻ seems to form the Frenkel excitons (J. Phys. Chem. Lett. 2013, 4, 11, 1788–1796). Upon radiation of X-rays, the heavy rare earth ions in the outer-shell are more vulnerable to be ionized to generate hot electrons for their higher X-ray absorption cross section. And the photogenerated electrons could further bring forth electron-hole pairs, which are bound by Coulomb force, leading to the formation of Frenkel excitons.

Response: The previously reported literature (High-resolution X-ray luminescence extension imaging. Nature 590, 410-415 (2021)) has been clearly verified the generation of F⁻ related Frenkel defects in fluoride NPs upon X-ray irradiation. We agree with the statement that “Upon radiation of X-rays, the heavy rare earth ions in the outer-shell are more vulnerable to be ionized to generate hot electrons for their higher X-ray absorption cross section”. This aligns with the mechanism presented in our manuscript: “When the X-ray photons interact with the heavy lanthanide ions, hot

electrons and deep holes are generated primarily through the photoelectric effect.”. Frenkel defects play a critical role as traps for capturing and depositing hot electrons, ultimately leading to persistent luminescence. It's important to note that the Frenkel defects formed within the fluoride nanocrystal structure differ from Frenkel excitons."

In this study, we highlight that small fluoride NPs possess a large surface-to-volume ratio, and then X-ray irradiation induces the formation of Frenkel defects predominantly on the NP surface. To verify whether incompletely coordinated surface fluoride ions indeed promote the formation of Frenkel defects upon ionizing radiation, we first employed the Density Functional Theory (DFT) to calculate the formation energies (E_f) of F^- ions with different coordination numbers. As illustrated in **Supplementary Fig S1a-b**, the F^- coordination number is reduced in the $NaLuF_4$ crystals with Na^+ vacancies. As shown in **Supplementary Fig S1c**, when dislocating F^- ions into interstitial sites at varying separation distances (0.5, 1.0, 1.5, and 2.0 Å), the E_f values were consistently smaller for F^- ions with fewer coordination numbers.

Furthermore, we compared the XEPL intensities of lanthanide-doped fluoride NPs before and after a wet chemical annealing process. Prior research (ACS Nano, 2018, 12, 4, 3623–3628) has established that wet chemical annealing can restore the arrangement of surface ions in fluoride NPs. Their results also revealed that the coordination numbers of F^- ions are reduced due to the presence of surface ion vacancy. To validate the impact of surface ion rearrangement on X-ray excited optical luminescence, we initially prepared $KLu_2F_7:15Tb$ NPs using a method similar to the reference mentioned above. Figure R4 demonstrates that the X-ray excited optical luminescence intensity significantly increased after wet chemical annealing. This enhancement is attributed to the rearrangement of surface ions and the subsequent reduction in surface defects. Unfortunately, the $KLu_2F_7:15Tb$ NPs do not exhibit XEPL (Figure R4d). We then investigated the X-ray excited optical luminescence and XEPL intensities of $NaLuF_4:15Tb$ NPs before and after annealing. As shown in **Supplementary Fig. S1d-g**, the X-ray excited optical luminescence intensity of the $NaLuF_4:15Tb$ NPs increased after annealing, while its XEPL intensity decreased. These results indicate that an incomplete surface ion arrangement can reduce X-ray

excited optical luminescence intensity while promoting an improvement in XEPL. Considering the XEPL mechanism, the enhanced XEPL in cases of incomplete surface ion arrangement can be attributed to an increase in F^- related Frenkel defects and subsequently an increase in the number of trapped electrons. These results manifested that incompletely coordinated surface F^- ions indeed facilitate the formation of Frenkel defects upon ionizing radiation.

Figure R4 XRD pattern (a) and TEM image (b) of the as-prepared $KLu_2F_7:15Tb$ NPs. (c) X-ray excited optical luminescence spectra of the $KLu_2F_7:15Tb$ NPs before and after wet chemical annealing. (d) XEPL spectra of the $KLu_2F_7:15Tb$ NPs before and after wet chemical annealing.

The following contents have been added in the revised manuscript.

Supplementary Fig. 1 (a) Crystal structure of perfect NaLuF₄. (b) Crystal structure of NaLuF₄ with Na⁺ vacancies. (c) Calculated Frenkel defect formation energies in these two crystal structures. TEM image (d) and XRD pattern (e) of the as-prepared NaLuF₄:15Tb NPs. X-ray excited optical luminescence (f) and XEPL (g) intensity variations of the NaLuF₄:15Tb NPs before and after annealing.

Supplementary Discussion. Density functional theory (DFT) was employed to calculate the formation energies (E_f) of anion Frenkel defects in NaLuF₄ crystals, both with and without Na⁺ vacancies. When dislocating F⁻ ions into interstitial sites at various separation distances (0.5, 1.0, 1.5, and 2.0 Å), the E_f values were consistently smaller for the F⁻ ions with fewer coordination numbers. Consequently, F⁻ ions with incomplete coordination on the surface are more inclined to form Frenkel defects compared to their fully coordinated counterparts.

Furthermore, it is noteworthy that the X-ray excited optical luminescence intensity of NaLuF₄:15Tb NPs increased following wet chemical annealing (ACS Nano, 2018, 12, 3623-3628), while its XEPL intensity decreased. This enhancement can be attributed to the rearrangement of surface ions and the subsequent reduction in surface defects. Considering the XEPL mechanism, the stronger XEPL observed in case of incomplete surface ion arrangement can be attributed to an increase in F⁻ related Frenkel defects and, consequently, a greater number of trapped electrons. These analyses provide further evidence that incompletely coordinated surface F⁻ ions indeed facilitate the formation of Frenkel defects upon exposure to ionizing radiation.

Reviewer 2:

Comment: The paper is focused on the behavior of X-ray excited persistent luminescence (XEPL) of lanthanide activated core and multishell fluoride NPs (NPs). The XEPL results enhanced by careful varying the architecture of the NPs, type of dopant and its concentration. The paper also describes an interesting investigation of the XEPL enhancement, mainly Dy³⁺ doped NPs, due to heterogeneous interfaces among core and shell layers, considering the generation of traps upon X-ray irradiation. An investigation concerning the XEPL for other lanthanide ions is also reported, evidencing a strong enhancement of the XEPL intensity for the core/shell/shell structure with respect to the core one. The high resolution 3D imaging investigation using a polymer based flexible film loaded with the NPs is also interesting for the scientific community. The paper deserves to be published in Nature Communication. I have only a couple of minor issues:

Response: We appreciate the reviewer for dedicating time to the thorough review of our manuscript and for their positive comment. We have diligently addressed the minor issues raised in the review, and these improvements have been incorporated into the revised manuscript. Your valuable input has contributed to enhancing the quality of our work.

Comment: the authors state that the dual heterogeneous interface promotes the generation of anion Frenkel defects at the heterogeneous interface: have the authors other experimental evidences of the increase of these defects? Is it possible at least to evaluate their concentrations in the core and core/shell (/shell) structures?

Response: In our original manuscript, we have already conducted X-ray Photoelectron Spectroscopy, geometric phase analysis, X-ray activated thermoluminescence, density functional theory calculations to verify that the dual heterogeneous interface promotes the generation of anion Frenkel defects at the heterogeneous interface. According to the reviewer's valuable comments and insightful suggestions, we further characterized the differences of captured electrons at trap states by electron paramagnetic resonance (EPR) spectroscopy for the

homogenous Lu@Lu/Dy@Lu and heterogeneous Y@Lu/Dy@Y core/shell/shell NPs. The g values (where g is a scaling factor accounting for the coupling between orbital and spin angular momentum) can be used to further verify the variations of the anion Frenkel defects. As shown in **Supplementary Fig. S22**, after cessation of X-ray excitation, the EPR signal intensity ($g = 1.8743$ and $g = 2.1163$) of the Y@Lu/Dy@Y NPs was stronger than that of the Lu@Lu/Dy@Lu counterparts. This result further confirm that the construction of a heterogeneous core@shell structure promotes the generation of traps upon X-ray irradiation.

The following content has been added in the revised manuscript.

After the cessation of X-ray excitation, the electron paramagnetic resonance (EPR) signal intensity ($g = 1.8743$, $g = 2.1163$) was stronger in the Y@Lu/Dy@Y NPs compared to the Lu@Lu/Dy@Lu counterparts (**Fig. S22**).

Supplementary Fig. 22 EPR spectra of the Lu@Lu/Dy@Lu and Y@Lu/Dy@Y NPs, before X-ray irradiation (a) and after X-ray irradiation (b). H, magnetic field.

The amount of X-ray induced Frenkel defects (n_F) can be expressed by the following equation [ref: Bollmann, W., Gorlich, P., Hauk, W. & Mothes, H. Ionic conduction of pure and doped CaF₂ and SrF₂ crystals. *Phys. Status Solidi A* **2**, 157-170 (1970)]:

$$n_F = \sqrt{N_I N_i} e^{-\frac{E_f}{2kT}}$$

where N_I and N_i are the number of F⁻ lattices and interstitial sites, respectively, k and T are Boltzmann constant and temperature, respectively. Thus a decrease of E_f can

lead to an evident increase of n_F . The E_f values for the F^- ions at different lattices in the same NP, and for dislocating the same F^- ion to different separation distances are much different. In our case, the high energy X-rays can dislocate F^- ions at different lattices to different interstitial sites, hence, it is unlikely to calculate the absolute Frenkel defects concentrations in the core and core/shell (/shell) structures. However, we can compare the relative Frenkel defects concentrations in similar cases. Considering this work highlights that the heterogeneous interface promotes the generation of anion Frenkel defects, the relative Frenkel defects concentrations in the Lu/Dy@Lu and Lu/Dy@Y core@shell NPs were evaluated. The calculated results were given in the following Table.

Sample	Distance (Å)	Ratio ($\times 10^{23}$)
Lu@Y : Lu@Lu	0.5	1.07
Lu@Y : Lu@Lu	0.7	1.17
Lu@Y : Lu@Lu	1	1.39
Lu@Y : Lu@Lu	1.5	7.64
Lu@Y : Lu@Lu	2	67.78

Comment: what is the spatial resolution that could be achieved in the 3D reconstruction? This is an interesting value to know in order to exploit this system in real technological applications.

Response: The spatial resolution achieved in the 3D reconstruction primarily relies on the components of the imaging system, including the camera (ORCA-Fusion BT), the lens (25 mm), and the optical setup. In this study, the parameters of the camera:

Pixel size: $6.5 \mu\text{m} \times 6.5 \mu\text{m}$;

Effective number of pixels: 2304×2304 ;

Full frame: $2304 \times 6.5 \mu\text{m} = 15 \text{ mm}$;

For the lens, the focal plane resolution is calculated as:

$1 / (160 \text{ lp/mm} \times 2) = 3.3 \mu\text{m}$;

Amplification: $A = \sqrt{2123} \div 15 = 3$;

The spatial resolution: $1000 \div (3 \times 6.5 \mu\text{m} \times 2) = 25.64 \text{ lp/mm}$.

Hence, the spatial resolution achieved in the 3D reconstruction model is

approximately 25.64 lp/mm.

The above calculated results and processes have been added in the revised manuscript and supporting information (See **Supplementary Fig. 35**).

Reviewer 3:

Comment: The investigation of NPs that show tunable x-ray excited persistent luminescence (XEPL) is certainly a very active and promising topic of applied nanoparticle research. The authors show here very interesting results by which they are able to enhance XEPL intensity signals largely by building NPs with a well-designed core-shell-shell structure. In addition, the authors also give good experiments to gain an insight into the mechanism which is behind the large intensity enhancement of their core-shell-shell NPs. In general, it can be stated that the manuscript is well structured and written very clearly. The characterization done on the NPs, as well as their application in sensing are done in high quality and the results are very convincing. Also the methods employed and how the results were obtained is described in a fully clear way and is well meeting high standards in the field.

Response: We appreciate the reviewer for dedicating valuable time to review our manuscript and for their positive comment.

Comment: Accordingly, in general the claims of this work are well substantiated. However, in order to appreciate further the enhancement of XEPL intensity reported here, it would be interesting to not just compare the outcome relative within the series of NPs produced by the authors. As the intensity is given in a.u. it is impossible to really appreciate the quality of the findings of this work, as one cannot compare the outcome to other work in the field. However, this is of central importance. Accordingly, one would either have to quantify the XEPL intensity in absolute terms compared to the x-ray intensity input or to compare under identical conditions with the performance of other, already published, NPs (systems). In summary, I would favour the publication of this contribution, but only once the claimed luminescence enhancement becomes clarified in absolute terms, so that it can really become compared to other comparable systems.

Response: We greatly appreciate the valuable comment provided by the reviewer. To quantitatively assess XEPL intensity in absolute terms, we compared the XEPL intensities of the Y@Lu/Gd/Tb@Y core@shell@shell NPs studied in this work with

those of commercial persistent phosphors, namely, $\text{CaAl}_2\text{O}_4:\text{Eu}/\text{Dy}$, $\text{SrAl}_2\text{O}_4:\text{Eu}/\text{Dy}$, and $\text{CaS}:\text{Eu}$, under various X-ray irradiation conditions. These commercial persistent phosphors were purchased from Xiucan Chemical and Andron Technologies company. The comparative XEPL intensities among these persistent phosphors are summarized in Figure X. With an increase in irradiation energy, the generation of secondary electrons also increases, resulting in enhanced XEPL. Due to differences in the variation trends among these systems, the relative XEPL intensities varied under different irradiation energies.

The following contents have been added in the revised manuscript.

The commercial $\text{CaAl}_2\text{O}_4:\text{Eu}/\text{Dy}$, $\text{SrAl}_2\text{O}_4:\text{Eu}/\text{Dy}$, and $\text{CaS}:\text{Eu}$ persistent phosphors were purchased from Xiucan Chemical and Andron Technologies company.

Under different X-ray irradiation conditions, the XEPL intensity of the $\text{Y}@\text{Lu}/\text{Gd}/\text{Tb}@\text{Y}$ core@shell@shell NPs was stronger than that of commercial counterparts ($\text{CaAl}_2\text{O}_4:\text{Eu}/\text{Dy}$, $\text{SrAl}_2\text{O}_4:\text{Eu}/\text{Dy}$, and $\text{CaS}:\text{Eu}$), and the relative intensities were summarized in Fig. S28.

Supplementary Fig. 28 Scanning electron microscope images of commercial persistent phosphors, CaS:Eu (**a**), CaAl₂O₄:Eu/Dy (**b**), SrAl₂O₄:Eu/Dy (**c**). EDX spectra of the CaS:Eu (**d**), CaAl₂O₄:Eu/Dy (**e**), SrAl₂O₄:Eu/Dy (**f**). XEPL spectra under different irradiation conditions, 20 KV (**g**), 30 KV (**h**), 40 KV (**i**), 50 KV (**j**). **k** Compared XEPL intensity ratios under different irradiation conditions.

Reviewer 4:

Comment: The quality of the experimental data is relatively poor and important data (x-ray diffraction, STEM images, chemical maps, EDX line scan) are missing for a large number of samples that are described in the manuscript. This prevents to properly compare the different samples. Favoring classical TEM instead of STEM is a methodological mistake especially when claiming differences between homogeneous and heterogeneous interfaces in core-shell materials. Finally, most of authors' claims are not supported by experimental evidence (see section 4) and most recent discoveries regarding core-shell NaREF₄ materials (see section 3) are not considered by the authors leading to misinterpretations (see section 4).

Response: Thank you for the insightful comment from the reviewer. We have included XRD, STEM-HAADF images, chemical maps, and EDX line scans for both the studied core@shell (such as, Lu/Gd/Dy@Lu, Lu/Gd/Dy@Y, Lu/Tb@Y) and core@shell@shell (such as, Lu@Lu/Dy@Lu, Lu@Lu/Dy@Y, Y@Lu/Dy@Y, Lu@Lu/Gd/Dy@Lu, Lu@Lu/Gd/Dy@Y, Y@Lu/Gd/Dy@Y) NPs in the revised manuscript and supporting information (See **Supplementary Fig. S5, S6, S7, S8, S11, S12, S17**). Furthermore, we have effectively addressed the concerns raised in sections 3 and 4. We sincerely hope that these revisions meet the reviewer's expectations.

Comment: The submitted manuscript appears to be an extension of authors' previous work (where XEPL was called XEA – X-ray excited afterglow), which was published in 2022 (Nature Communications 2022, 13, 5739). The heterogeneous aspect, supposed to be the main novelty of the submitted manuscript, was already described (to a certain extent) in Nature Communications (2022, 13, 5739). This is also the case for the mechanistic investigation (see details below), which is exactly the same as reported in Nature Communications (2022, 13, 5739). This combined to the fact that the conclusions are not supported by the experimental data, the significance of the manuscript is very limited.

Response: We appreciate the reviewer for valuable comment, and we have conducted a thorough comparison between the content of our current work and our previous

publication in Nature Communications 2022, 13: 5739. In our prior research, our primary focus was on enhancing the X-ray excited photoluminescence (XEPL) of lanthanide activators through the incorporation of interstitial Na⁺ ions within the nanocrystal structure. This approach led to the development of core@shell NPs displaying distinctive time-dependent afterglow color evolution. Additionally, we explored the creation of core@shell@shell structures to achieve multimode time-dependent multicolored afterglow emissions. In contrast, our current study is dedicated to enhancing XEPL of lanthanide activators by designing dual interfaces within a heterogeneous core@shell@shell structure. The core objective is to enable low-dose, high-resolution delayed XEPL-based 3D imaging, which represents a departure from the objectives outlined in our previous work.

For a more comprehensive understanding of the differences between our present research and our prior work, we provide a detailed comparison below.

- (1) **Methods for XEPL Enhancement:** Our previous work primarily focused on enhancing XEPL intensities of lanthanide activators (Pr, Tb, Dy, Sm) through the incorporation of interstitial Na⁺ ions within the nanocrystal structure. In contrast, the present study concentrates on elevating XEPL intensities of lanthanide activators (Pr, Tb, Dy, Er, Tm, Gd) by designing dual interfaces within a heterogeneous core@shell@shell structure.
- (2) **Chemical Compositions:** In our previous work, our primary compositions consisted of NaLuF₄:15Gd/15Tb core NPs, core@shell NPs including NaLuF₄:15Gd/15Tb@NaLuF₄:10Gd/0.5Sm, NaLuF₄:15Gd/0.5Sm@NaLuF₄:15Gd/0.5Dy, NaLuF₄:15Tb@NaLuF₄:15Gd/0.5Pr core@shell NPs, and core@shell@shell NPs such as NaLuF₄:15Gd/15Tb@NaLuF₄:15Gd/10Ce/0.5Sm@NaGdF₄:49Yb/1Tm. While the core@shell@shell structure exhibited heterogeneity in the outer shell, the shells were primarily designed to facilitate downshifting and upconversion emissions rather than XEPL. The XEPL, in this case, originated primarily from the NaLuF₄:15Gd/15Tb core layer. Consequently, the core@shell@shell NPs investigated in our previous work are entirely unrelated to the concept of

enhancing XEPL by designing dual interfaces in a heterogeneous core@shell@shell structure. In our prior work, the NaYF₄@NaLuF₄:15Gd/15Tb structure only emerged once, and its purpose was to verify that the XEPL of Tb ions could be improved by increasing the [Na]/[RE] ratio in the shell layer. The two compared core/shell NPs are nearly unrelated to the concept of enhancing XEPL through the design of dual interfaces in a heterogeneous core@shell@shell structure.

- (3) **Mechanistic Investigations:** In our prior research, we employed techniques such as inductively coupled plasma-optical emission spectroscopy (ICP-OES) and Rietveld XRD refinement to confirm the presence of interstitial Na⁺ ions within the nanocrystal structure. Subsequently, we used X-ray photoelectron spectroscopy (XPS), X-ray activated thermoluminescence (TL) glow curve analysis, and Density Functional Theory (DFT) calculations to assess the impact of these interstitial Na⁺ ions on XEPL intensity. In contrast, in the present work, our mechanistic investigations followed a distinct path. We utilized STEM-HAADF, mapping, EDX line scans, and geometric phase analysis (GPA) to elucidate the differences between homogeneous and heterogeneous core@shell@shell structures. Subsequently, we employed XPS, X-ray activated TL glow curve analysis, DFT calculations, and electron paramagnetic resonance (EPR) to confirm the influence of heterogeneous core@shell@shell structures on XEPL intensity. While some characterization techniques were shared between the two studies, the specific content and objectives of these investigations were entirely different. Furthermore, in our prior work (Nature Communications 2022, 13, 5739), we highlighted the role of interstitial Na⁺ ions in promoting the generation of anion Frenkel defects upon X-ray irradiation within the fluoride nanocrystal structure. In contrast, in the present work, we emphasize that "XEPL intensity could be significantly enhanced by constructing a core@shell@shell structure with dual heterogeneous interfaces." This novel core@shell@shell design serves two critical purposes: it passivates surface quenchers, reducing non-radiative relaxation possibilities, and it lowers the interfacial Frenkel defect

formation energy, resulting in increased trap concentration due to the presence of misfit strain at the heterogeneous interfaces. These insights represent a substantial departure from our previous work and underscore the unique contributions of the present study.

(4) **Applications:** Notably, the applications explored in our previous work and the present study were markedly different. In our prior research, we focused on the fascinating realm of time-dependent multicolor evolution. In contrast, the current study is dedicated to the intriguing domain of low-dose, high-resolution delayed XEPL-based 3D imaging. Significantly, the present study made a pivotal discovery. When examining a real-time X-ray image of an electronic watch, we observed the presence of several bright spots. However, these conspicuous spots were entirely absent in the delayed X-ray image. This novel finding underscores a new and crucial advantage of delayed X-ray imaging over real-time X-ray imaging, which has been documented for the first time. This discovery has profound implications and enhances our understanding of the capabilities of delayed XEPL-based imaging techniques, setting it apart from our previous work, which concentrated on entirely distinct applications.

Comment: The manuscript is poorly written and extremely hard to understand even for someone in the field. The choice of samples' abbreviations makes the reading very unpleasant and the reader constantly has to refer to Table 1.

Response: We appreciate the reviewer's helpful comment, and we have carefully revised the whole manuscript for better comfortable reading. Additionally, we have revised the abbreviations for the samples to make them more intuitive and consistent. For example, we have abbreviated "NaLuF₄:20Gd/2Dy@NaLuF₄" as "Lu/Gd/Dy@Lu," "NaLuF₄:2Dy@NaLuF₄" as "Lu/Dy@Lu," and "NaYF₄@NaLuF₄:2Dy@NaYF₄" as Y@Lu/Dy@Y. Furthermore, in the revised manuscript, we have included the notation: "Unless otherwise specified, we prepared the emissive and inert layers using [Na]/[RE] ratios of 10 and 2.5, respectively." This addition helps clarify the experimental conditions for the readers. It is worth noting

that Gd doping was consistently maintained at 20 mol%, while Dy doping was at 2 mol%, unless explicitly stated otherwise. Therefore, we have omitted doping concentrations from the abbreviations.

We believe these revisions address the reviewer's concerns and contribute to a clearer and more reader-friendly presentation of our work.

Comment: Figures' captions, are not self-explanatory and the reader must dig into the text to understand the figures. It is absolutely impossible to understand the figures with the chosen abbreviations.

Response: According to the reviewer's helpful comment, the Figures' captions in this manuscript have been revised. The abbreviations are presented as below.

Fig. 1 Compared XEPL properties of Dy doped fluoride NPs with different structures.

Fig. 2 Compared XEPL properties of lanthanide activators doped in the homogeneous and heterogeneous core@shell@shell structures.

Fig. 3 Analysis of interfacial structure and energy transfer in the heterogeneous core@shell@shell NPs.

Fig. 4 XEPL characterizations of more lanthanide activators doped in the heterogeneous core@shell@shell structure.

Fig. 5 Delayed XEPL-based 3D imaging.

Comment: Authors botched supplementary figures' captions.

Response: According to the reviewer's helpful comment, all the supplementary Figures' captions in the supplementary information have been revised.

Comment: In the main text and figures' captions, authors alternate between abbreviations and full names. Authors should stick to only one type throughout the text.

Response: According to the reviewer's helpful comment, the abbreviations of the studied samples were used throughout the text.

Comment: The manuscript does not appropriately refer to the existing literature. In particular, authors completely ignored intermixing problems observed with NaREF₄ core-shell materials (e.g. Chem. Mater. 2015, 27, 8375; Small 2021, 17, 2104441; J. Am. Chem. Soc. 2023, <https://doi.org/10.1021/jacs.3c03019>).

Response: We sincerely appreciate the valuable comment provided by the reviewer. After carefully reviewing the suggested references, we agree that cation intermixing likely occurs during the growth of shells in our core@shell and core@shell@shell NPs. In the referenced literature, it is elucidated that the dissolution of seed core NPs is a prominent mechanism contributing to cation intermixing in core@shell NPs. Furthermore, these studies confirm that the substantial core NP dissolution and subsequent reformation in synthetic solvents at high temperatures are strongly influenced by the number density of the particles regardless of the NP size, composition, heterostructure, and phase. In our experimental setup, both the core@shell and core@shell@shell NPs were prepared under identical experimental parameters, especially with regard to number density. The primary variation lies in the choice of lanthanide activators. Therefore, we anticipate that the issue of cation intermixing should be similar between Lu/Gd/Dy@Lu and Lu/Gd/Dy@Y core@shell NPs, as well as among Lu@Lu/Dy@Lu, Lu@Lu/Dy@Y, and Y@Lu/Dy@Y core@shell@shell NPs.

In the suggested reference (J. Am. Chem. Soc. 2023, <https://doi.org/10.1021/jacs.3c03019>), it was demonstrated that the stability of seed core NP with a diameter of 25 nm significantly increased when their number density reached $8.57 \times 10^{13} \text{ mL}^{-1}$. The disintegration of NPs is regulated by a dissolution equilibrium

where RE³⁺ represents rare earth ions. The equilibrium constant of NaREF₄ NPs, namely, the solubility product K_{sp}, can be described by

$$K_{\text{sp}} = [\text{Na}^+][\text{RE}^{3+}][\text{F}^-]^4$$

Importantly, in this context, they also confirmed that the excess sodium ions can effectively protect the structural integrity of the core NPs.

In our case, the mean core size in the Lu@Lu/Dy@Lu, Lu@Lu/Dy@Y and Y@Lu/Dy@Y core@shell@shell NPs were about 25 nm as well. We determined the number density of the core NPs using the method outlined in the referenced study.

The volume of a single NP (ligand free) is

$$V_{NP} = \frac{4}{3} \times \left(\frac{d}{2}\right)^3 \times \pi = \frac{4}{3} \times \left(\frac{25}{2}\right)^3 \times 3.14 = 8.18 \times 10^3 \text{ nm}^3,$$

while the average mass of a single NP (ligand free) is

$$m_{NP} = \rho_{NaYF_4} \times V_{UCNP} = 4.35 \times 10^{-21} \times 8.18 \times 10^3 = 3.56 \times 10^{-17} \text{ g}.$$

Considering the mass loss of OA ligands, the average mass of initial OA-coated single

$$\text{NP is } m_{NP-OA} = \frac{3.56 \times 10^{-17}}{80\%} = 4.45 \times 10^{-17} \text{ g}.$$

As a result, the “molecular weight” of initial OA-coated single NP is

$$M_{NP-OA} = m_{NP-OA} \times N_A = 4.45 \times 10^{-17} \times 6.02 \times 10^{23} = 2.68 \times 10^7 \text{ g/mol}.$$

$$\text{The total molar amount NPs is } \frac{0.0744}{2.68 \times 10^7} = 2.78 \times 10^{-9} \text{ mol}.$$

At this time, the total number of NPs is

$$N_{NP-OA} = 2.78 \times 10^{-9} \times 6.02 \times 10^{23} = 1.672 \times 10^{15}.$$

Finally, the number density is about $\frac{1.672 \times 10^{15}}{20} = 8.36 \times 10^{13} \text{ mL}^{-1}$, which is similar to

the stable case observed in the above reference.

Furthermore, in our case, the [Na]/[RE] ratio for the active shell layer is 10, which is significantly higher than necessary. Consequently, the core NPs studied in our experiments exhibit a relatively high level of stability, leading to a minimal occurrence of cation intermixing.

To further investigate whether the issue of cation intermixing affects the conclusion that 'heterogeneous interfaces benefit the improvement of XEPL intensity,' we use core NPs with a low number density ($2.09 \times 10^{13} \text{ mL}^{-1}$) and a low Na^+

concentration in the shell layer ($[Na]/[RE] = 2.5$) to prepare Lu/Dy@Lu and Lu/Dy@Y core@shell NPs. These two core@shell NPs share the same core size and exhibit similar shell thickness (Fig. S20 a-c). The HAADF image and mapping results confirmed the formation of core@shell structure (Fig. S20d-h). As depicted in Fig. S20i, it is evident that the XEPL intensity of heterogeneous Lu/Dy@Y core@shell NPs was substantially stronger than that of homogeneous Lu/Dy@Lu core@shell NPs. Hence, our findings indicate that the issue of cation intermixing does not alter the conclusion that “heterogeneous interfaces contribute to the enhancement of XEPL intensity” in this study.

The following contents including the suggested references have been added in the revised manuscript and supporting information.

Despite the possible presence of cation intermixing during shell growth (Chem. Mater. 2015, 27, 8375; Small 2021, 17, 2104441; J. Am. Chem. Soc. 2023, 145, 32, 17621–17631), the XEPL intensity of core@shell NPs with a heterogeneous structure remained stronger than that of their homogeneous counterparts (Fig. S20).

Supplementary Fig. 20 TEM images of the Lu/Dy core (a), Lu/Dy@Lu (b), Lu/Dy@Y (c) core@shell NPs. HAADF image (d), element mapping results (e, f), mixed HAADF image and Lu signal (g), mixed HAADF image and Y signal (h), of the Lu/Dy@Y core@shell NPs.

The cation intermixing issue during shell growth varies with the number density of core NPs (J. Am. Chem. Soc. 2023, 145, 32, 17621–17631). We used core NPs with a low number density ($2.09 \times 10^{13} \text{ mL}^{-1}$) and a low Na^+ concentration in the shell layer ($[\text{Na}]/[\text{RE}] = 2.5$) to prepare Lu/Dy@Lu and Lu/Dy@Y core@shell NPs. These two core@shell NPs share the same core size and exhibit similar shell thickness. It is evident that the XEPL intensity of heterogeneous Lu/Dy@Y core@shell NPs was substantially stronger than that of homogeneous Lu/Dy@Lu core@shell NPs. This observation underscores that the issue of cation intermixing does not alter the

conclusion that heterogeneous interfaces contribute significantly to the enhancement of XEPL intensity.

Comment: "The XEPL intensity of the Y_{2.5}@Dy₁₀@Y_{2.5} NPs was approximately 40.9 times higher than that of Dy_{2.5} NPs, where the subscript numbers represent their fractions in the composition."

Comparing core-shell NPs with pure core NPs is not a fair comparison due to significantly enhanced surface quenching in the latter case. What did you expect?

Response: We appreciate the reviewer's observation regarding the comparison between core@shell@shell and pure core NPs. Surface quenching is indeed a critical factor influencing luminescence intensity of NPs. In our study, we aimed to highlight the significance of the core@shell@shell structure in enhancing luminescence. We acknowledge that pure core NPs have inherent surface quenchers, but in our case, the interfacial layer thickness of the NaYF₄@NaLuF₄:20Gd/2Dy@NaYF₄ core@shell@shell NPs (approximately 7 nm) was significantly smaller than that of NaLuF₄:20Gd/2Dy core NPs (approximately 35 nm). Even with a homogeneous shell up to 14 nm, the XEPL intensity was only enhanced approximately 2.3 times. It is important to note that the formula of NaLuF₄:20Gd/2Dy represents a common structure in the XEPL field. Therefore, we emphasize that the substantial improvement in XEPL intensity of Dy activators can be achieved by strategically designing a core@shell@shell structure with dual heterogeneous interfaces.

The original sentence has been revised to "By restricting the Dy activators to the interfacial layer of the Y@Lu/Gd/Dy@Y heterogeneous core/shell/shell NPs, we observed a remarkable 40.9-fold enhancement in XEPL intensity compared to that of the general Lu/Gd/Dy core NPs."

Comment: "The XEPL intensity of the Y_{2.5}@Dy₁₀@Y_{2.5} NPs was approximately 40.9 times higher than that of Dy_{2.5} NPs, where the subscript numbers represent their fractions in the composition."

The underlined text is absolutely not clear and does not mean anything. It also seems to be wrong because when reading the text it seems that 2.5 and 10 refer to the [Na]/[RE] ratio.

Response: Thank you for pointing out the need for clarity in our sample abbreviations. The samples' abbreviations have been revised accordingly, for examples, the "NaLuF₄:20Gd/2Dy" was abbreviated as "Lu/Gd/Dy, [Na]/[RE] = 2.5" or "Lu/Gd/Dy, [Na]/[RE] = 10", the "NaLuF₄:20Gd/2Dy@NaLuF₄" was abbreviated as "Lu/Gd/Dy@Lu", the "NaLuF₄:20Gd/2Dy@NaYF₄" was abbreviated as "Lu/Gd/Dy@Y", the "NaYF₄@NaLuF₄:20Gd/2Dy@NaYF₄" was abbreviated as "Y@Lu/Gd/Dy@Y". The notation of "Unless otherwise specified, we prepared the emissive and inert layers using [Na]/[RE] ratios of 10 and 2.5, respectively." has been added in the revised manuscript. Gd doping was consistently maintained at 20 mol%, while Dy doping was at 2 mol% unless specified otherwise. So doping concentrations were omitted in the abbreviations.

Comment: "All these NPs were well indexed to a pure hexagonal phase without any impurity peaks (Fig. S2)".

How do the authors explain that Bragg peaks of 35 nm core NPs (Dy_{2.5}) are as sharp (or very likely even sharper) as Bragg peaks of larger (62-65 nm) core-shell NPs (Dy₁₀@Lu_{2.5}, Dy₁₀@Y_{2.5}, Y_{2.5}@Dy₁₀@Y_{2.5})?

Response: Thank you for pointing out this mistake, and we have measured the XRD patterns of these samples carefully again. The revised XRD patterns (**Supplementary Fig. 3**) were added in the supporting information.

Supplementary Fig. 3 XRD patterns of the Lu/Gd/Dy ([Na]/[RE] = 2.5), Lu/Gd/Dy ([Na]/[RE] = 10), Lu/Gd/Dy@Lu, Lu/Gd/Dy@Y, Y@Lu/Gd/Dy@Y NPs. The bars represent the standard data of hexagonal NaLuF₄ (JCPDS No. 270726).

Comment: "The Y element was observed in the Dy₁₀@Y_{2.5} core/shell NPs (Fig. S3d), but not in the Dy₁₀ core layer (Fig. S3b), indicating the formation of a core/shell structure."

A basic large area EDX analysis performed on a large number of particles (supplementary figure 3) does not prove the formation of core-shell structures. Why would you like to find Y in pure core NPs (Dy₁₀) that do not contain Y (NaLuF₄:20Gd/2Dy according to Table 1)? This sentence makes no sense.

Response: Based on the reviewer's insightful comment, the high-angle annular dark-field (HAADF) image, EDX line scan and element mapping analysis were used to confirm the formation of Lu/Gd/Dy@Y core@shell structure. As shown in **Supplementary Fig. S6**, the Lu was mainly distributed in the core while the Y was primarily located in the shell layer. The original sentence was corrected to "The high-angle annular dark-field (HAADF) images, EDX line scans and element mapping results confirmed the formation of Lu/Gd/Dy@Lu (Fig. S5) and Lu/Gd/Dy@Y (Fig. S6) core@shell as well as Y@Lu/Gd/Dy@Y (Fig. 1b and S7, S8) core@shell@shell structures."

Supplementary Fig. 6 HAADF image (a), mixed Lu and Y signals (b), EDX line scan (c), element mapping results (d, e, f), mixed HAADF image and Gd signal (g), mixed HAADF image and Lu signal (h), mixed HAADF image and Y signal (i), of the Lu/Gd/Dy@Y core@shell NPs.

Comment: "The core/shell/shell structure of the Y_{2.5}@Dy₁₀@Y_{2.5} NPs was clearly verified by the high angle annular dark-field (HAADF) TEM image, which is sensitive to the atomic numbers^{29,30} (Fig. 1b)"

Scanning transmission electron microscopy is not just sensitive to the chemical composition but also to the thickness. The STEM image presented in Figure 1b is given with false colors. The reviewer wants to see the corresponding raw STEM image. Performing HAADF-STEM is important and must be performed on ALL samples reported in the manuscript including low magnification to assess the homogeneity of the synthesized NPs. This is not the case and authors mostly included

TEM images, which are not as informative as STEM images when dealing with core-shell NPs.

Response: We appreciate the reviewer's valuable comment. The raw STEM image corresponding to the Figure 1b has been added in the revised supporting information (**Supplementary Fig. S7**). Moreover, the HAADF-STEM and low magnification TEM images of the studied core@shell (such as, Lu/Gd/Dy@Lu, Lu/Gd/Dy@Y) and core@shell@shell (such as, Lu@Lu/Dy@Lu, Lu@Lu/Dy@Y, Y@Lu/Dy@Y, Lu@Lu/Gd/Dy@Lu, Lu@Lu/Gd/Dy@Y, Y@Lu/Gd/Dy@Y) NPs were provided in the revised supporting information (See **Supplementary Fig. S5, S6, S7, S8, S12, S17**).

Supplementary Fig. 7 HAADF image of the Y@Lu/Gd/Dy@Y NPs corresponding to the image presented in Fig. 1b.

Comment: "EDX line scan and mapping analysis revealed that Lu was primarily distributed in the interlayer and Y was located in the core and the outer shell (Fig. 1c), which were consistent with the nominal compositions of the core/shell/shell structure."

The reviewer would like to see the STEM image from which the chemical maps presented in figure 1c have been obtained. The STEM image must be given at least in the supplementary information. The reviewer would like to see the Y and Gd maps

(individual channels), which should be added in the supplementary information. Finally, the Lu chemical map makes no sense for a core-shell-shell nanoparticle. Indeed, Y_{2.5}@Dy₁₀@Y_{2.5} refers to NaYF₄@NaLuF₄:Gd:Dy@NaYF₄. This means that Lu (and Gd) are confined (according to the authors) in the interlayer. Therefore, it is impossible to get nearly no Lu counts in the central region of the particles as demonstrated with completely black areas. Can the authors clarify? Did the authors perform post-processing to clean up images? The line scan analysis is also not in agreement with the formation of core-shell-shell NPs because the Y signal has a very similar magnitude compared to the Lu signal on the sides of the particle, which makes no sense because the external shell is supposed to be a pure Y shell. This can be a sign of cation intermixing, which was not considered by the authors.

Response: Following the valuable comment from the reviewer, we recharacterized the Y@Lu/Gd/Dy@Y core@shell@shell NPs using EDX line scans and elemental mapping. The STEM image for the chemical maps in Fig. 1c, along with the individual Y and Gd maps (**Supplementary Fig. 8**), has been added to the revised supplementary information. The new line scan results align with the formation of the core@shell@shell structure (Fig. 1c).

Fig. 1 Compared XEPL properties of Dy doped fluoride NPs with different structures. **a** Schematic illustration for the influence of shell layer on the passivation of surface quenchers and the generation of Frenkel defects. The core NP contains many surface quenchers, and the incompletely coordinated surface F⁻ ions facilitate the formation of Frenkel defects upon X-ray irradiation. The inert shell serves to passivate the surface quenchers. A homogenous core@shell structure hinders Frenkel defects formation, whereas the heterogeneous core@shell structure with a heterogeneous interface promotes it. HAADF image (**b**), EDX line scan and mapping results (**c**) of the Y@Lu/Gd/Dy@Y NPs. XEPL spectra (**d**), integral XEPL intensities (**e**), XEPL decay curves (**f**), and digital photographs (**g**) of the studied NPs at different time delays.

Supplementary Fig. 8 HAADF image (a), and the corresponding element mapping results of the Y@Lu/Gd/Dy@Y NPs.

The raw STEM image for the chemical maps presented in original Figure 1c as well as the raw Lu map were shown in the Figure R5. To improve visibility, we adjusted the brightness of the element mapping image, which is almost the same with the original image shown Figure 1c and does not influence the analysis of actual element distribution. In the HAADF image, it is noticeable that the exposed NPs on the copper grid exhibit varying shapes. For the NP-2, the exposed interfacial layer is very thin, resulting in a similar contrast between the core and shell layer. Consequently, the Lu signal is recorded in both the core and interfacial layer. Conversely, for the NP-1, the exposed interfacial layer is thicker, resulting in a significant contrast between the core and shell layers. Consequently, the Lu signal in the core layer is much weaker than in the interfacial layer.

While EDX line scans are commonly used to analyze the elemental composition of solid NPs, it is crucial to consider their limitations. In the case of core@shell NPs, the elements present in the shell layer can often be detected in the core region during an EDX line scan, but with relatively weak signal intensities. Furthermore, the dissolution of core NPs within a very thin layer might not result in significant signal variations, making it challenging to accurately confirm cation intermixing through EDX line scans. In the suggested references of Chem. Mater. 2015, 27, 8375 and J. Am. Chem. Soc. 2023, <https://doi.org/10.1021/jacs.3c03019>, to study the cation intermixing phenomenon, the EDX line scan were not used. In the suggested reference 'Small 2021, 17, 2104441,' the authors emphasize that “To summarize, raw

compositions obtained from EDX spectra cannot be used to reveal the exact local chemical composition of core@shell HNCs without further processing because the electron beam propagates through the shell and core yielding an averaged chemical composition along the electron-beam direction.” Hence, the EDX line scans were not used to study the cation intermixing in this study.

Figure R5 HAADF image, Lu mapping result and the schematic illustrations of the NP shape appeared on the copper grid.

Comment: "Increasing the [Na]/[RE] ratio in the precursor solution from 2.5 to 10 resulted in a 6.8-fold increase in the XEPL intensity (Fig. 1d-e)"

The authors did not quantify the [Na]/[RE] ratio. This should be done for all reported samples in the manuscript.

Response: Following the valuable comment from the reviewer, the [Na]/[RE] ratio of the Lu/Gd/Dy ([Na]/[RE] = 2.5), Lu/Gd/Dy ([Na]/[RE] = 10), Lu/Gd/Dy@Lu, Lu/Gd/Dy@Y, Y@Lu/Gd/Dy@Y NPs were measured by inductively coupled plasma-optical emission spectroscopy and provided in **Supplementary Fig. 4f**.

Supplementary Fig. 4 EDX spectra of the Lu/Gd/Dy ([Na]/[RE] = 2.5) (a), Lu/Gd/Dy ([Na]/[RE] = 10) (b), Lu/Gd/Dy@Lu (c), Lu/Gd/Dy@Y (d), Y@Lu/Gd/Dy@Y (e) NPs. (f) [Na]/[RE] ratios measured by inductively coupled plasma-optical emission spectroscopy. [RE] = [Lu] + [Gd] + [Dy].

Comment: "Interestingly, the XEPL intensity of the Dy10@Y2.5 NPs was much stronger than that of the Dy10@Lu2.5 NPs with similar core sizes and shell

thicknesses, indicating that the heterogeneous interface produced an improvement in the XEPL intensity."

This is pure speculation. The authors only presented basic TEM images (supplementary figure 1) instead of STEM images. Chemical maps and EDX line scans should be performed to validate the core-shell formation. The Gd signal can be utilized (due to its relatively high concentration of 20%) even for the homogeneous NPs.

Response: Following the valuable comment from the reviewer, the HAADF images, EDX line scans and element mapping of the Lu/Gd/Dy@Lu (**Supplementary Fig. 5**) and Lu/Gd/Dy@Y core@shell (**Supplementary Fig. 6**) NPs were characterized and added in the revised supporting information.

The high-angle annular dark-field (HAADF) images, EDX line scans and element mapping results confirmed the formation of Lu/Gd/Dy@Lu (Fig. S5) and Lu/Gd/Dy@Y (Fig. S6) core@shell as well as Y@Lu/Gd/Dy@Y (Fig. 1b and S7, S8) core@shell@shell structures.

Supplementary Fig. 5 HAADF image (a), element mapping results (b, c), mixed HAADF image and Gd signal (d), mixed Gd and Lu signals (e), EDX line scan (f), of the Lu/Gd/Dy@Lu core@shell NPs.

Supplementary Fig. 6 HAADF image (a), mixed Lu and Y signals (b), EDX line scan (c), element mapping results (d, e, f), mixed HAADF image and Gd signal (g), mixed HAADF image and Lu signal (h), mixed HAADF image and Y signal (i), of the Lu/Gd/Dy@Y core@shell NPs.

Comment: "Confirming the enhancement of XEPL by designing comparable nanostructures"

For this second set of samples, the authors added Dy in the interlayer and not in the core compared to the first set of samples for which Dy was in the core. Moreover, Gd has been added in the first set of samples but not in the second. Therefore, authors do not compare absolutely equivalent homogeneous (set #2) and heterogeneous (set #1) NPs. It is also obvious from figure 2 that the quality of the synthesized homogeneous (figures 2a-b) and heterogeneous (figures 2d-f) NPs is not equivalent. See for instance figures 2a (bottom) and 2e or figures 2b and 2f. Equally important, all corresponding

STEM images, chemical maps, and EDX line scans are missing for samples presented in Figure 2. Those are of major interest especially when dealing with core-shell structures.

Response: We appreciate the reviewer's valuable comment. The first set of samples are Lu/Gd/Dy ([Na]/[RE] = 2.5), Lu/Gd/Dy ([Na]/[RE] = 10), Lu/Gd/Dy@Lu, Lu/Gd/Dy@Y, and Y@Lu/Gd/Dy@Y NPs, which are used to reveal that the Y@Lu/Gd/Dy@Y core@shell@shell NPs with dual heterogeneous interfaces exhibited stronger XEPL intensity than the other four kinds of NPs. The second set of samples are Lu@Lu/Dy@Lu, Lu@Lu/Dy@Y, Y@Lu/Dy@Y core@shell@shell NPs, which are further used to confirm the advantage of the heterogeneous interfaces. Considering the excitation energy can be efficiently transferred from Gd³⁺ to Dy³⁺ activators, the Gd ions were not incorporated in the interlayer to avoid their influences on the XEPL intensity variations. It should be noted that we do not compare the XEPL intensities between the two sets of samples.

To further confirm the heterogeneous core@shell@shell NPs presented stronger XEPL intensity than that of homogeneous ones even with doping Gd³⁺ ions in the interfacial layer, the Lu@Lu/Gd/Dy@Lu, Lu@Lu/Gd/Dy@Y, Y@Lu/Gd/Dy@Y core@shell@shell NPs were prepared and studied. As shown in Fig. S16-17, the TEM and HAADF images revealed that these NPs exhibited similar core size and shell thickness. As anticipated, the Y@Lu/Gd/Dy@Y core@shell@shell NPs with dual heterogeneous interfaces exhibited stronger XEPL intensity than those of Lu@Lu/Gd/Dy@Lu and Lu@Lu/Gd/Dy@Y NPs (Fig. S18).

We feel sorry about that it is almost impossible to prepare uniform pure NaLuF₄ core NPs without doping under the same experimental conditions for that of pure NaYF₄ core NPs, even we tried many times to prepare a series of NaLuF₄ core NPs. In this study, the Lu@Lu/Gd/Dy@Lu, Lu@Lu/Gd/Dy@Y, Y@Lu/Gd/Dy@Y core@shell@shell NPs possessed very similar mean sizes of core, first shell and second shell. Considering the XEPL intensity is the integral intensities of all NPs of a sample, not a single NP, it is feasible to compare the XEPL intensities among these core@shell@shell NPs. Moreover, we compared many sets of samples to reveal the

heterogeneous interfaces benefit the improvement of XEPL intensity, such as “Lu/Gd/Dy@Lu and Lu/Gd/Dy@Y core@shell NPs”, “Lu@Lu/Dy@Lu, Lu@Lu/Dy@Y, and Y@Lu/Dy@Y core@shell@shell NPs (d=5nm)”, “Lu@Lu/Dy@Lu, Lu@Lu/Dy@Y, and Y@Lu/Dy@Y core@shell@shell NPs (d=9nm)”, “Lu@Lu/Gd/Dy@Lu, Lu@Lu/Gd/Dy@Y, and Y@Lu/Gd/Tb@Y”, “Lu@Lu/Tb@Lu, Lu@Lu/Tb@Y, and Y@Lu/Tb@Y”. It should be emphasized that the homogeneous Lu/Gd/Dy@Lu and heterogeneous Lu/Gd/Dy@Y NPs possessed almost the same core and shell size distributions (**Supplementary Fig. 2**). As a result, the conclusion of “XEPL intensity can be improved by constructing heterogeneous interface” is reliable.

According to the reviewer’s helpful comment, all corresponding STEM images, chemical maps, and EDX line scans for samples presented in Figure 2 were provided in the revised supporting information (See **Supplementary Fig. S5, S6, S7, S8, S12, S17**).

The following content has been added in the revised manuscript and supporting information.

Interestingly, even with the incorporation of Gd³⁺ ions in the interfacial layer, the Y@Lu/Gd/Dy@Y NPs continued to exhibit the highest XEPL intensity (Fig. S16, S17, S18).

Supplementary Fig. 16 TEM images of the NaLuF₄ core (a), Lu@Lu/Gd/Dy core@shell NPs (b), and corresponding histogram of shell thickness distributions (c). TEM images of the NaYF₄ core (d), Y@Lu/Gd/Dy core@shell NPs (e), and corresponding histogram of shell thickness distributions (f). TEM images of the Lu@Lu/Gd/Dy@Lu (g), Lu@Lu/Gd/Dy@Y (h), Y@Lu/Gd/Dy@Y (i) core@shell@shell NPs.

Supplementary Fig. 17 HAADF images and element mapping results of the Lu@Lu/Gd/Dy@Lu (a) and Lu@Lu/Gd/Dy@Y (b) core@shell@shell NPs.

Supplementary Fig. 18 XEPL spectra of the Lu@Lu/Gd/Dy@Lu, Lu@Lu/Gd/Dy@Y, Y@Lu/Gd/Dy@Y core@shell@shell NPs.

Comment: The mechanistic investigation described in the manuscript is exactly the same as the one described in Nature Communications 2022, 13, 5739. Are there differences?

Response: XEPL in lanthanide doped fluoride NPs originates from the release of deposited electrons in the traps formed by the F⁻ related Frenkel defects after the cessation of X-rays. This mechanism has been well studied in the previous literature (High-resolution X-ray luminescence extension imaging. Nature 590, 410-415 (2021)). However, it is rarely reported about how to promote the generation of F⁻ related Frenkel defects and then enhance the XEPL intensity. In the previous work (Nature Communications 2022, 13, 5739), we reported that the formed interstitial Na⁺ ions in the fluoride nanocrystal structure could promote the generation of anion Frenkel defects upon X-ray irradiation. In the present work, we emphasize that “the XEPL intensity could be greatly enhanced by constructing a core@shell@shell structure with a dual heterogeneous interface. The employed core@shell@shell structure serves two purposes: passivating the surface quenchers to reduce non-radiative relaxation possibilities, and lowering the interfacial Frenkel defect formation energy, leading to increased trap concentration due to the presence of misfit strain at the heterogeneous interfaces”, which are almost completely different from the previous work. Moreover, in the present work, the STEM-HAADF, mapping, EDX line scan and geometric phase analysis (GPA) were employed to reveal the differences between the homogeneous and heterogeneous core/shell/shell structures. Then the XPS, X-ray activated thermoluminescence (TL) glow curves, DFT calculations and electron paramagnetic resonance (EPR) were further used to confirm the influence of heterogeneous core/shell/shell structures on the XEPL intensity, which are also greatly different from the previous work.

Comment: the authors never considered cation intermixing and the modification of real concentrations including non-homogeneous distributions and concentration

gradients (Small 2021, 17, 2104441; J. Am. Chem. Soc. 2023, <https://doi.org/10.1021/jacs.3c03019>) despite the fact that their experimental data do not allow to exclude such a phenomenon.

Response: Thanks for the reviewer’s insightful comment. All of the NPs studied in our research were prepared using a similar co-precipitation method. Consequently, the issues related to cation intermixing and the modification of real concentrations, including non-homogeneous distributions and concentration gradients, are indeed similar among the compared NPs. We have conducted additional experiments to address this issue, and the detailed experimental results have been added along with the previous Comment “The manuscript does not appropriately refer to the existing literature. In particular, authors completely ignored intermixing problems observed with NaREF₄ core-shell materials (e.g. Chem. Mater. 2015, 27, 8375; Small 2021, 17, 2104441; J. Am. Chem. Soc. 2023, <https://doi.org/10.1021/jacs.3c03019>).”

Comment: Figure 3d: the corresponding indexation of the FFT is missing.

Response: According to the reviewer’s kind comment, the corresponding indexation of the FFT has been added in the revised image (Fig. 3d).

Fig. 3 Analysis of interfacial structure and energy transfer in the heterogeneous core@shell@shell NPs. d FFT pattern of a single Y@Lu/Dy@Y NP.

Comment: “To confirm the presence of lattice distortion in the heterogeneous interface, a fast Fourier transform (FFT) pattern analysis was conducted. The FFT pattern revealed two distinct sets of diffraction spots corresponding to the core and the

interlayer of a single NP, indicating a difference in lattice constants between the two regions (Fig. 3d). This observation suggests the presence of lattice distortion at the interface³³. A geometric phase analysis (GPA) of the high-resolution TEM image of a single core/shell/shell NP was employed to visualize the strain distribution^{22,34}."

Pure speculation. You do not have the necessary resolution for such measurements (just look at the FFT presented in figure 3d). Let's assume authors managed to distinguish 2 sets of diffraction spots, this means authors should have seen 2 sets of Bragg peaks by X-ray diffraction. This is not the case because of the overlap due to the small size of the coherent domains. Broad peaks in X-ray diffraction also means diffuse spots in electron diffraction. What are the calculated values of the extracted cell parameters and what are the corresponding errors and standard deviations?

Response: Following the valuable comment from the reviewer, an HRTEM image with better resolution was used to perform FFT (See revised Fig. 3d). Two distinct sets of diffraction spots were still recorded in the FFT pattern. And two Zoom-in patterns were provided as well.

The cell parameters were calculated by the following equation

$$\frac{1}{d_{hkl}^2} = \frac{4(h^2 + hk + k^2)}{3a^2} + \frac{l^2}{c^2}$$

The zone axis for the diffraction pattern presented in Fig. 3d is [001], and the corresponding crystal planes are (hk0). In this particular case, the c cannot be calculated based on this FFT pattern. For the $(\bar{3}00)$ plane, the d values corresponding to the core and interfacial layer are 1.713 Å and 1.731 Å, respectively. Then the calculated a values are $a=5.934 \pm 0.012$ Å and $a=5.996 \pm 0.010$ Å, respectively.

A high-resolution TEM image allows us to reveal the microscopic structure of a NP, enabling us to obtain independent FFT patterns of the core and shell within a single core@shell NP. In bulk systems, the hexagonal NaYF₄ ($a=b=5.96$ Å, $c=3.53$ Å) and NaLuF₄ ($a=b=5.901$ Å, $c=3.453$ Å) phases exhibit different crystal lattice parameters and interplanar spacing (d) values. Therefore, it is reasonable to expect distinct FFT patterns for the core and interfacial layer in heterogeneous

core@shell@shell NPs. This phenomenon has also been reported in a previous study titled “Accurate Control of Core–Shell Upconversion NPs through Anisotropic Strain Engineering, *Adv. Funct. Mater.* 2019, 1903295”. In contrast to high-resolution TEM imaging, XRD patterns provide a macroscopic and statistical overview of all nanoparticles in a sample. The hexagonal NaYF₄ and NaLuF₄ phases generally yield similar XRD patterns, with subtle shifts due to the different ionic radii of Y (1.159 Å) and Lu (1.117 Å). Consequently, the two sets of Bragg peaks are not always distinguishable within the same XRD pattern, primarily due to peak broadening and the resolution limits of XRD instruments.

However, differences in XRD patterns between Y@Y/Dy@Y and Y@Lu/Dy@Y NPs are still observed. The slow-scan XRD patterns of these NPs are presented in Supplementary Fig. 23. In the XRD pattern of core@shell NPs, each 2θ value corresponds to a statistical value associated with a crystal face in both the core and shell. When the crystal symmetry between the core and shell is different, the relative variations in d values for different crystal faces will also be different, leading to distinct variations in 2θ values. Notably, the difference in 2θ values between the heterogeneous Y@Lu/Dy@Y core@shell@shell NPs and homogeneous Y@Y/Dy@Y core@shell@shell NPs was significant (Fig. S23), indicating substantial differences in d ratios ($d_{(112)}/d_{(220)}$, $d_{(202)}/d_{(310)}$) between the NaYF₄ and NaLuF₄ crystals. Consequently, the Fast Fourier Transform (FFT) patterns corresponding to the core (NaYF₄) and the interlayer (NaLuF₄) of a single nanoparticle are distinct, resulting in two distinct sets of diffraction spots.

The following content has been added in the revised manuscript and supporting information.

The distinctions observed in the slow-scan XRD patterns between the Y@Y/Dy@Y and Y@Lu/Dy@Y NPs (Fig. S24) manifest substantial variations in interplanar spacing values and their corresponding ratios between the NaYF₄ and NaLuF₄ crystals.

Supplementary Fig. 24 Low-scan XRD patterns of the heterogeneous Y@Lu/Dy@Y and homogeneous Y@Y/Dy@Y core@shell@shell NPs.

Comment: What are the results when performing delayed XEPL-based 3D imaging with homogeneous core-shell NPs that are strictly equivalent to their heterogeneous counterparts?

Response: According to the reviewer’s kind comment, the delayed XEPL-based 3D imaging with homogeneous Lu@Lu/Gd/Tb@Lu core@shell@shell NPs that are strictly equivalent to their heterogeneous counterparts was performed. As shown in Fig.S35, under the same X-ray irradiation dose rate of 4.5 $\mu\text{Gy/s}$, the delayed XEPL-based image exhibited significantly lower clarity compared to the image obtained using heterogeneous Y@Lu/Gd/Tb@Y core@shell@shell NPs (**Supplementary Fig. S36**).

Supplementary Fig. 36 Delayed XEPL based image of the electronic watch from different directions, Lu/Tb core NPs (a), homogeneous Lu@Lu/Gd/Tb@Lu core@shell@shell NPs (b).

These contents have been added in the revised manuscript and supporting information.

REVIEWER COMMENTS

Reviewer #1 (Remarks to the Author):

The majority of the mentioned problems have been resolved, but there still remains a small fraction that requires further resolution.

1. Some details should be supplemented: 1) What software is used to achieve 3D model reconstruction in this article? 2) In the preparation of PDMS film with scintillators, which crosslinking agents were used to achieve the polymerization reaction of PDMS prepolymers at 70 °C?

2. The ratio between rare earth elements is overly idealized. During synthesis, the conversion of raw materials can hardly be 100%. Please use ICP-OES or ICP-MS or other measurements to determine the true proportions of rare earth elements in these core@shell and core@shell@shell NPs. And maybe, the details like a Table or Notes should be added, especially in Supplementary Fig. 4f.

3. The whole paper exhibits the flexible three-dimensional imaging using a flexible scintillating film. But the manuscript did not tell us how flexible the film is, please provide some more visual mechanical properties like tensile strength, compressibility, resilience and essential fracture properties of this flexible film.

Reviewer #2 (Remarks to the Author):

The revised paper can be accepted for publication.

Reviewer #3 (Remarks to the Author):

The authors have successfully replied to the points of criticism raised from my side. I think that technically it is a sound paper. Of course, its degree of novelty is a bit limited, as many relevant aspects of this publication are already shown and discussed in their Nature Comm. paper from 2022. However, that is more an editorial decision than for a reviewer, as the paper itself is clearly sound and well-written.

Reviewer #4 (Remarks to the Author):

The authors made an effort to improve the clarity of their manuscript, which is appreciated. Not all comments were properly addressed and the reviewer still disagrees with the explanations given and in particular regarding the formation of heterogeneous core-shell-shell nanoparticles (NPs). Although the reviewer agrees with the authors that differences regarding the XEPL properties do exist between the "homogeneous" and "heterogeneous" samples, the exact reason is not obvious. Experimental data (and in particular the requested STEM images) clearly indicate that the formation of an authentic heterogeneous core-shell-shell structure (Y@Lu/Gd/Dy@Y) is not true. In fact the synthesized heterogeneous core-shell-shell NPs are a mixture of different types of NPs. Because the experimental evidence do not support the conclusions, the reviewer still considers the manuscript is not suitable for Nature Communications. The reviewer would like to emphasize important points:

- 1) What a pity that low magnification (i.e. several hundreds of NPs per image) STEM images are not given as requested. This would really help to assess the homogeneity of the synthesized samples.
- 2) The reviewer would like to emphasize once again that the STEM data clearly indicate that the "heterogeneous" NPs (Y@Lu/Gd/Dy@Y), in fact, contain four different types of particles as highlighted in Fig. R1 (please description attached with Fig. R1)

We clearly see the interest of HAADF-STEM images. Indeed, compared to Figure 2f, it is now very easy to prove the existence of a non-homogeneous sample, which was initially claimed as heterogeneous

core-shell-shell NPs. Such a trend is further confirmed by Supplementary Fig. 17 with chemical maps. Those chemical maps do not exhibit obvious chemical patterns regarding the distribution of the different chemical elements.

3) Fig. 1c: EDX chemical maps and line scan profiles have, very likely, been extracted from the first type of NPs (see Fig. R1). Here we nicely distinguish the expected chemical distribution pattern for core-shell-shell NPs. But it also seems that such an expected chemical pattern is not true for other particles (see Supplementary Fig. 17 and Fig. R2). Also, how is it possible to get EDX signals at zero for both Y and Lu (see Fig. R2). Such a weird behavior is also observed for core-shell NPs (Lu/Gd/Dy@Y) as shown in Supplementary Fig. 6c.

4) Missing HAADF-STEM images for the claimed heterogeneous core-shell-shell NPs described in Supplementary Fig. 14d (Y), 14e (Y@Lu/Dy), and 14 (Y@Lu/Dy@Y). The reviewer can already tell, only based on given TEM images, that very likely no heterogeneous core-shell-shell have been formed (massive changes observed in the shape distribution and faceting). Chemical maps should be given to definitely decide.

5) The reviewer still disagrees with the authors regarding the fact that electron and x-ray diffraction prove the formation of heterogeneous core-shell-shell structures. Both electron and x-ray diffraction are sensitive to the size of the coherent domains. Peaks broadening in the case of x-ray diffraction is related to the sample (size of the coherent domains and microstrain) and the instrument (x-ray source, optics etc.). Separated electron diffraction spots should be correlated with separated Bragg peaks. We come back to point #2, the authors selected the first type of NPs. The authors explain themselves the difference (p. 49):

"In contrast to high-resolution TEM imaging, XRD patterns provide a macroscopic and statistical overview of all nanoparticles in a sample."

All types (first, second, third and fourth) of NPs are indeed analyzed by x-ray diffraction. So analyzing and selecting only one type of NPs is not a proof that all synthesized NPs are heterogeneous core-shell-shell.

5) The authors indicate slow-scan (Supplementary information) and low-scan (main text) XRD. What is it?

6) The authors indicate (p. 49):

"When the crystal symmetry between the core and shell is different, the relative variations in d values for different crystal faces will also be different, leading to distinct variations in 2θ values."

The crystal symmetry is exactly the same for the core and shell (hexagonal). Thus, the given explanation makes no sense.

Response to reviewer's comments

We greatly appreciate the reviewers' insightful comments which were very helpful for the improvement of our revised manuscript. In response to the valuable comments raised by the referees, we provide point-by-point responses along with the modifications (marked in blue) made in the revised manuscript.

Reviewer 1:

Comment: Some details should be supplemented: 1) What software is used to achieve 3D model reconstruction in this article? 2) In the preparation of PDMS film with scintillators, which crosslinking agents were used to achieve the polymerization reaction of PDMS prepolymers at 70 °C?

Response: Thanks for the reviewer's valuable comment. The 3D model structures were reconstructed using the Maxon Cinema 4D R26 software, which has been added in the experimental section of the revised supporting information.

The PDMS was purchased from Dow Corning Company and consisted of two bottles: one containing the prepolymer and the other holding the crosslinker. The bottles did not display the specific chemical compositions, and similar to the reference in Nature Communications, 2019, 10:1391, the compositions of the crosslinking agent was not provided either.

Comment: The ratio between rare earth elements is overly idealized. During synthesis, the conversion of raw materials can hardly be 100%. Please use ICP-OES or ICP-MS or other measurements to determine the true proportions of rare earth elements in these core@shell and core@shell@shell NPs. And maybe, the details like a Table or Notes should be added, especially in Supplementary Fig. 4f.

Response: We appreciate the insightful comment from the reviewer. The numbers presented in the product formulas, such as "20/2" in the NaLuF₄:20Gd/2Dy NPs, were nominal rare earth doping concentrations, not actual ones. The actual rare earth doping concentrations were measured by ICP-OES, which have been added in the revised Supplementary Fig. 4f.

f

Sample	[Lu]	[Gd]	[Dy]	[Na] / [RE]
Lu/Gd/Dy (2.5)	75.82	22.35	1.83	1.091
Lu/Gd/Dy (10)	76.78	21.47	1.75	1.324
Lu/Gd/Dy@Lu	88.01	11.09	0.90	1.145
Lu/Gd/Dy@Y	75.62	22.54	1.84	1.151
Y@Lu/Gd/Dy@Y	76.61	21.68	1.71	1.127

Supplementary Fig. 4 (f) [Na]/[RE] and activators doping ratios measured by inductively coupled plasma-optical emission spectroscopy.

Comment: The whole paper exhibits the flexible three-dimensional imaging using a flexible scintillating film. But the manuscript did not tell us how flexible the film is, please provide some more visul mechanical properties like tensile strength, compressibility, resilience and essential fracture properities of this flexible film.

Response: According to the reviewer’s valuable comment, more visul mechanical properties of the film were provided in Supplementary Fig. 29. The as-prepared film can be bent up to approximately 180 degrees, and stretched more than 200%.

The following content has been added in the revised manuscript.

The flexible film exhibited excellent flexibility to be conformable, with the ability to bend up to approximately 180 degrees and stretch more than 200% (Fig. 5a and S29).

Supplementary Fig. 29 Photographs of the original (a), folded (b), stretched (c), and bent (d) Y@Lu/Gd/Tb@Y core@shell@shell NPs integrated flexible film.

Reviewer 4:

Comment: What a pity that low magnification (i.e. several hundreds of NPs per image) STEM images are not given as requested. This would really help to assess the homogeneity of the synthesized samples.

Response: We deeply appreciate the insightful feedback from the reviewer. We did indeed capture low-magnification STEM images of the as-prepared core@shell NPs, however, as presented in Figure R2, it is relatively hard to distinguish the core@shell@shell structure based on the low-magnification STEM image. Additionally, we employed a relatively low concentration of NPs during our TEM characterizations to mitigate potential issues related to NP stacking on the TEM copper grid. High NP concentration can adversely affect the mapping results and lead to data inaccuracies.

Figure R2 Low magnification STEM image of the Y@Lu/Gd/Dy@Y core@shell@shell NPs.

In many previous literatures, high-magnification STEM images were used to clearly reveal the core@shell or core@multi-shells structures, such as *Nature Nanotechnology*, 2018, 13, 941–946; *Chem. Mater.* 2019, 31, 15, 5608–5615; *Nano Lett.* 2021, 21, 4838–4844; *ACS Photonics*, 2022, 9, 758–764; *Nature Communications*, 2020, 11, 1174; *Chem. Mater.* 2013, 25, 106–112; *Chem. Mater.* 2016, 28, 7, 2295–2300.

We will discuss the issue of sample homogeneity in the next comment.

Comment: The reviewer would like to emphasize once again that the STEM data clearly indicate that the "heterogeneous" NPs (Y@Lu/Gd/Dy@Y), in fact, contain four different types of particles as highlighted in Fig. R1 (please description attached with Fig. R1). We clearly see the interest of HAADF-STEM images. Indeed, compared to Figure 2f, it is now very easy to prove the existence of a non-homogeneous sample, which was initially claimed as heterogeneous core-shell-shell NPs. Such a trend is further confirmed by Supplementary Fig. 17 with chemical maps. Those chemical maps do not exhibit obvious chemical patterns regarding the distribution of the different chemical elements.

Response: We appreciate the insightful comment from the reviewer. The as-prepared NaYF₄ core, Y@Lu/Gd/Dy core@shell and Y@Lu/Gd/Dy@Y core@shell@shell NPs displayed hexagonal prism shape, as shown in Supplementary Figure 2. The two-dimensional shape observed in the TEM image can vary based on the random deposition of nanoparticles on the copper grid. As schematically illustrated in Figure R3, the shape may differ when capturing a same nanoparticle from different angles or directions. Consequently, in Supplementary Figure 7 (Figure R3b-c), we observed four different types of nanoparticles. The result observed in Supplementary Fig. 17 is attributed to the similar reason.

Figure R3 Schematic illustration of the projected two-dimensional image of a NP with a hexagonal prism shape from various angles or directions.

Actually, this phenomenon were observed in many previous literatures, such as Nano Lett. 2012, 12, 2852–2858; J. Phys. Chem. Lett. 2017, 8, 5099–5104; small 2017, 1701479; Chem. Mater. 2019, 31, 15, 5608–5615.

To help assessing the distribution of chemical elements, we have marked a few particles in Supplementary Figure 17.

Supplementary Fig. 17 HAADF images and element mapping results of the Lu@Lu/Gd/Dy@Lu (a), Lu@Lu/Gd/Dy@Y (b) and Y@Lu/Gd/Dy@Y (c) core@shell@shell NPs.

To further validate that the enhanced XEPL intensity is mainly attributed to the generation of heterogeneous interface, but not the shape, we prepared sphere-like Lu@Lu/Tb@Lu, Lu@Lu/Tb@Y, and Y@Lu/Tb@Y core@shell@shell NPs with shell thickness of approximately 2 nm using a thermal decomposition method (as previously described in Nano Lett. 2019, 19, 3878–3885). As shown in Figure R4, we can observe that the XEPL intensity of the Y@Lu/Tb@Y NPs is stronger than that of Lu@Lu/Tb@Y NPs, and both are stronger than that of Lu@Lu/Tb@Lu NPs. These findings provide further confirmation that the XEPL intensity of lanthanide-doped fluoride nanoparticles with a sphere-like shape can be significantly enhanced by the construction of a heterogeneous interface.

Figure R4 TEM images of the NaLuF₄ (**a, top**), Lu@Lu/Tb (**a, bottom**), Lu@Lu/Tb@Lu (**b**), Lu@Lu/Tb@Y (**c**), NaYF₄ (**d**), Y@Lu/Tb (**e**), Y@Lu/Tb@Y (**f**). **g** HAADF image and corresponding element mapping results of the Y@Lu/Tb@Y core@shell@shell NPs. **h** XEPL spectra of the Lu@Lu/Tb@Lu, Lu@Lu/Tb@Y, Y@Lu/Tb@Y core@shell@shell NPs.

We sincerely hope that these explanations meet the reviewer's expectations.

Comment: Fig. 1c: EDX chemical maps and line scan profiles have, very likely, been extracted from the first type of NPs (see Fig. R1). Here we nicely distinguish the expected chemical distribution pattern for core-shell-shell NPs. But it also seems that such an expected chemical pattern is not true for other particles (see Supplementary Fig. 17 and Fig. R2). Also, how is it possible to get EDX signals at zero for both Y and Lu (see Fig. R2). Such a weird behavior is also observed for core-shell NPs (Lu/Gd/Dy@Y) as shown in Supplementary Fig. 6c.

Response: We appreciate the reviewer for their valuable comment. The two-dimensional shape observed in the TEM image can vary based on the random deposition of nanoparticles on the copper grid, leading to differences in the chemical patterns. Nevertheless, the core@shell@shell structures of the NPs with various two-dimensional shapes were still distinctly confirmed by the STEM image.

The EDX characterization conditions: Dwell time 4.00 μ s, 4.2s /frame, Screen current 78 pA. The original data has been provided along with the manuscript.

As depicted in Figure R5, the results of element mapping and EDX line scans for the NPs with different two-dimensional shapes remained consistent with the core@shell@shell structure. It is important to note that the EDX line scan profiles, which extend along a line through the center of a single core@shell@shell nanoparticle, represent averaged values along the electron-beam direction. This implies that the entire volume along the electron trajectory contributes to the detected signal. Consequently, both the Y and Lu signals were recorded in the core layer for the type 3 nanoparticle (Figure R5b-c). From this perspective, it is more accurate and clear to reveal the element distributions in the core@shell@shell NPs based on the type 1 nanoparticle.

Figure R5 (a) HAADF image and element mapping results of the Y@Lu/Gd/Dy@Y core@shell@shell NPs. (b) EDX line scan of a single NP (type 2, type 3). (c) Schematic illustration of the element distributions of the core@shell@shell NPs from different directions or angles.

In the EDX line scan presented in Fig. 1c, the EDX signals at the marked '1', '2' and '3' points were not zero (Figure R6). All of these signals were higher than the

baseline. Notably, at point '2', when the pattern is enlarged, it becomes evident that the signal for Y is stronger than that for Lu.

Figure R6 Zoom-in EDX pattern corresponding to Fig. 1c.

When the pattern presented in Supplementary Figure 6c is enlarged, similar results are observed (Figure R7).

Figure R7 Zoom-in EDX pattern corresponding to Supplementary Fig. 6c.

The EDX-line scan and mapping results can only qualitatively represent the overall element variation. As stated in the reference 'Small 2021, 17, 2104441,' it is emphasized that “raw compositions obtained from EDX spectra cannot be used to reveal the exact local chemical composition of core@shell HNCs without further processing because the electron beam propagates through the shell and core yielding an averaged chemical composition along the electron-beam direction.” Element mapping and EDX line scans are sensitive to environmental variations, such as thermal and vibration, and they require a relatively long scanning time. Under these conditions, signals might exhibit slight fluctuations. However, the overall trends in element variation observed in our study are consistent with the nominal element distributions in the core@shell@shell NPs.

Comment: 4) Missing HAADF-STEM images for the claimed heterogeneous core-shell-shell NPs described in Supplementary Fig. 14d (Y), 14e (Y@Lu/Dy), and 14 (Y@Lu/Dy@Y). The reviewer can already tell, only based on given TEM images, that very likely no heterogeneous core-shell-shell have been formed (massive changes observed in the shape distribution and faceting). Chemical maps should be given to definitely decide.

Response: According to the reviewer's helpful comment, the HAADF-STEM images for the NPs described in Supplementary Fig. 14d (Y), 14e (Y@Lu/Dy), and 14 (Y@Lu/Dy@Y) were characterized.

These NPs displayed hexagonal prisms shape. The original TEM images for the (Y) and (Y@Lu/Dy@Y) presented in Supplementary Figure 14 were the top of hexagonal prism, while that of (Y@Lu/Dy) was the side of hexagonal prism, which were not attributed to the changed of the shape. To maintain consistency of the TEM image, the top of hexagonal prism for all the three NPs were provided.

Supplementary Fig. 14 TEM images of the NaLuF₄ (a, top), Lu@Lu/Dy (a, bottom), Lu@Lu/Dy@Lu (b), Lu@Lu/Dy@Y (c), NaYF₄ (d), Y@Lu/Dy (e), Y@Lu/Dy@Y (f). HAADF images of the NaYF₄ (g), Y@Lu/Dy (h), Y@Lu/Dy@Y (i). j HAADF image and corresponding element mapping results of the Y@Lu/Dy@Y core@shell@shell NPs. The interlayer thickness is about 9 nm.

Comment: The reviewer still disagrees with the authors regarding the fact that

electron and x-ray diffraction prove the formation of heterogeneous core-shell-shell structures. Both electron and x-ray diffraction are sensitive to the size of the coherent domains. Peaks broadening in the case of x-ray diffraction is related to the sample (size of the coherent domains and microstrain) and the instrument (x-ray source, optics etc.). Separated electron diffraction spots should be correlated with separated Bragg peaks. We come back to point #2, the authors selected the first type of NPs. The authors explain themselves the difference (p. 49):

" In contrast to high-resolution TEM imaging, XRD patterns provide a macroscopic and statistical overview of all nanoparticles in a sample."

All types (first, second, third and fourth) of NPs are indeed analyzed by x-ray diffraction. So analyzing and selecting only one type of NPs is not a proof that all synthesized NPs are heterogeneous core-shell-shell.

Response: We appreciate the reviewer for insightful comment. XRD patterns provide a macroscopic and statistical overview of all NPs in a sample, encompassing both the core and shells. Consequently, it can be challenging to discern the microstructures of the core and shells separately solely based on the XRD pattern. However, the high-resolution TEM image allows us to analyze the microstructure of the core and shells independently.

As we discussed in a previous comment, the different shapes of the Y@Lu/Gd/Dy@Y core@shell@shell nanoparticles are attributed to the projection of these nanoparticles from different angles or directions, rather than indicating different types of nanoparticles. Therefore, it is reasonable to analyze the high-resolution TEM image of a random single nanoparticle. In fact, as demonstrated in Figure R8, for the described type 2 nanoparticle, the separation of electron diffraction spots was also observed.

Figure R8 HAADF, high-resolution TEM image and the corresponding FFT pattern of a single NP (type 2).

Comment: The authors indicate slow-scan (Supplementary information) and low-scan (main text) XRD. What is it?

Response: We appreciate the reviewer's helpful comment. In order to emphasize the distinctions between the Y@Lu/Dy@Y and Y@Y/Dy@Y core@shell@shell NPs, we reduced the step size during the XRD measurement from 0.02 to 0.002. The term 'low-scan' has been corrected to 'slow-scan' in the revised caption of Supplementary Fig. 24. To enhance clarity, we have also included information about the step size in the caption of Supplementary Fig. 24.

Supplementary Fig. 24 Slow-scan XRD patterns of the heterogeneous Y@Lu/Dy@Y and homogeneous Y@Y/Dy@Y core@shell@shell NPs. Step-size, 0.002.

Comment: The authors indicate (p. 49): "When the crystal symmetry between the core and shell is different, the relative variations in d values for different crystal faces will also be different, leading to distinct variations in 2θ values."

The crystal symmetry is exactly the same for the core and shell (hexagonal). Thus, the given explanation makes no sense.

Response: We appreciate the helpful comment from the reviewer. In bulk systems, the hexagonal NaYF₄ (a=b=5.96 Å, c=3.53 Å) and NaLuF₄ (a=b=5.901 Å, c=3.453 Å) phases exhibit different crystal lattice parameters and interplanar spacing (d) values. We apologize for the inaccurate description of different crystal symmetry between the core and shell. We would like to emphasize the c/a ratio is different between NaYF₄ and NaLuF₄. Consequently, the relative variations in d values for different crystal faces will also be different, leading to different variations in 2θ values. The other explanations, such as "Notably, the difference in 2θ values between the heterogeneous Y@Lu/Dy@Y core@shell@shell NPs and homogeneous Y@Y/Dy@Y core@shell@shell NPs was significant (Fig. S23), indicating substantial differences in d ratios ($d_{(112)}/d_{(220)}$, $d_{(202)}/d_{(310)}$) between the NaYF₄ and NaLuF₄ crystals.

Consequently, the Fast Fourier Transform (FFT) patterns corresponding to the core (NaYF_4) and the interlayer (NaLuF_4) of a single nanoparticle are distinct, resulting in two distinct sets of diffraction spots.” remain reasonable.